# Rab7-dependent regulation of goblet cell protein CLCA1 modulates gastrointestinal homeostasis

Preksha Gaur[1], Yesheswini Rajendran[1], Bhagyashree Srivastava[2], Manasvini Markandey[3], Vered Fishbain-Yoskovitz[4], Gayatree Mohapatra[4], Aamir Suhail[5], Shikha Chaudhary[6], Shaifali Tyagi[7], Subhash Chandra Yadav[6], Amit Kumar Pandey[7], Yifat Merbl[4], Avinash Bajaj[1], Vineet Ahuja[3], Chittur Srikanth[1]*

[1]Laboratory of Gut Inflammation and Infection Biology, Regional Centre for Biotechnology, Faridabad, India; [2]Departmnet of Bioscience and Biotechnology, Banasthali Vidyapith, Aliyabad, India; [3]Department of Gastroenterology, All India Institute of Medical Sciences, Delhi, India; [4]Department of Immunology, Weizmann Institute of Science, Rehovot, Israel; [5]Gene Lay Institute of Immunology and Inflammation, Brigham and Women's Hospital, Massachusetts General Hospital and Harvard Medical School, Boston, United States; [6]Department of Anatomy, All India Institute of Medical Sciences, New Delhi, India; [7]Vaccine and Infectious Disease Research Center, Translational Health Science and Technology Institute, Faridabad, India

*For correspondence: cvsrikanth@rcb.res.in

Competing interest: The authors declare that no competing interests exist.

**Abstract** Inflammation in ulcerative colitis is typically restricted to the mucosal layer of distal gut. Disrupted mucus barrier, coupled with microbial dysbiosis, has been reported to occur prior to the onset of inflammation. Here, we show the involvement of vesicular trafficking protein Rab7 in regulating the colonic mucus system. We identified a lowered Rab7 expression in goblet cells of colon during human and murine colitis. In vivo Rab7 knocked down mice (Rab7$^{KD}$) displayed a compromised mucus layer, increased microbial permeability, and depleted gut microbiota with enhanced susceptibility to dextran sodium-sulfate induced colitis. These abnormalities emerged owing to altered mucus composition, as revealed by mucus proteomics, with increased expression of mucin protease chloride channel accessory 1 (CLCA1). Mechanistically, Rab7 maintained optimal CLCA1 levels by controlling its lysosomal degradation, a process that was dysregulated during colitis. Overall, our work establishes a role for Rab7-dependent control of CLCA1 secretion required for maintaining mucosal homeostasis.

## eLife assessment

This is an **important** study for understanding the pathogenesis of ulcerative colitis. It **convincingly** demonstrates reduced levels of the vesicular trafficking protein Rab7 in ulcerative colitis and Crohn's disease, leading to altered levels of calcium-activated chloride channel regulator 1 (CLCA1) and subsequent mucin dysregulation, highlighting Rab7's significance in gut homeostasis maintenance. The article advances the field as it provides insights into a novel regulatory pathway implicated in ulcerative colitis, potentially paving the way for the development of targeted therapeutic interventions.

## Introduction

A close association of gut with diverse microbial antigens demand for well-regulated mechanisms for tolerance and homeostasis. On a cellular level, intestinal epithelium and local immune cells execute diverse regulatory mechanisms to maintain this intestinal homeostasis (*Maloy and Powrie, 2011*). Goblet cells, a specialized epithelial cell type, participate by secreting mucins that form mucus layer, functioning as the first line of defense against the microbial population (*Johansson and Hansson, 2016*). Impairment in mucus layer has been observed before intestinal inflammation in ulcerative colitis (UC), one of the two major forms of inflammatory bowel disease (IBD) (*Boltin et al., 2013*; *Johansson et al., 2010*). Active UC patients exhibit reduced number of goblet cells and a compromised mucus layer (*Singh et al., 2022*). This highlights the importance of goblet cell secretory function and mucus layer in UC pathogenesis. Two stratified layers of mucus exist in colon: a loose outer layer permeable to the luminal flora and a dense adherent sterile inner layer (*Atuma et al., 2001*). The core structure of mucus is formed of Muc2. Several other known protein constituents of mucus layer are FCGBP, CLCA1, TFF3, and ZG16 (*Hansson and Johansson, 2010*). Chloride channel accessory 1 (CLCA1) is a metalloprotease exclusively secreted by goblet cells and forms one of the major non-mucin components of mucus (*Nyström et al., 2018*). It has Muc2 cleaving properties and thus is involved in intestinal mucus dynamics and homeostasis (*Nyström et al., 2019*). In view of its ability to protect intestinal epithelium and prevent infections and inflammation, the underlying cause of aspects related to mucus layer alterations during intestinal inflammation needs investigation.

In a recent report focusing on epithelial-immunocyte crosstalk, our group demonstrated a critical role of SENP7, a deSUMOylase in IBD pathophysiology (*Suhail et al., 2019*). Interestingly, Rab7, among the several other Rab GTPases identified to physically interact with SENP7, was observed in murine model of IBD. Rab7 protein participates in vesicular transport of a cell, influencing protein turnover, secretion, autophagy, and intracellular pathogen survival (*Deffieu et al., 2021*; *Gutierrez et al., 2004*; *Mohapatra et al., 2018*; *Vanlandingham and Ceresa, 2009*; *Yap et al., 2018*). Each of these functions is arguably essential in intestinal homeostasis, yet these have not been studied in detail in the context of IBD. With all the knowledge provided earlier, it is apparent that Rab proteins may have a role in IBD pathogenesis. But surprisingly, not a lot is known about these proteins and their specific role in IBD. In light of this information, in the present study, we aimed to systematically look at the role of Rab7 in UC pathogenesis. Our data suggests that dysregulated Rab7 during colitis modulates the secretions from goblet cells thus triggering inflammation in the gut.

## Results

### Rab7 expression is reduced during murine and human colitis

To understand the possible role of Rab7 in gut homeostasis and its relevance to colitis, we utilized dextran sulfate sodium (DSS) murine model. The animals were fed with 2.5% DSS in their drinking water, and their body weights were regularly monitored. After 7 d of DSS treatment (hereafter DSS mice), they were euthanized and various tissues were harvested for analysis. Compared to control mice which were given normal drinking water, the DSS mice displayed discernible signs of inflammation, including reduced body weight, decreased colon length, and various histopathological features (as depicted previously by *Suhail et al., 2019*). We examined Rab7 expression in inflamed intestine harvested from DSS-colitis mice. A significant reduction in Rab7 protein was seen in tissue lysates of distal colon from DSS mice compared to healthy control (*Figure 1A*). Immunohistochemistry (IHC) of colon sections revealed the ubiquitous presence of Rab7 protein over whole tissue with intensified expression in crypts and other zones of epithelium in healthy mice which got significantly lessened in DSS mice (*Figure 1—figure supplement 1A*, inset). To further understand the dynamics of the Rab7 expression change during the course of disease development, DSS treatment was given for different time durations, that is, 3 d (DSS 3), 5 d (DSS 5), or 7 d (DSS 7) (*Figure 1—figure supplement 1B*). Increased duration of DSS treatment resulted in a gradual decrease in mice body weight (*Figure 1—figure supplement 1C*) and colon length (*Figure 1—figure supplement 1D*) in DSS 7 and DSS 5 mice when compared with DSS 3 mice. DSS 5 and DSS 7 mice also displayed increased splenomegaly (*Figure 1—figure supplement 1E*), mirroring severe inflammation seen during human IBD. While whole colonic tissue lysate and crypts (isolated from colons of mice) showed upregulated expression of Rab7 at the early inflammation states (DSS 3–5) (*Figure 1—figure supplement 1F*), interestingly,

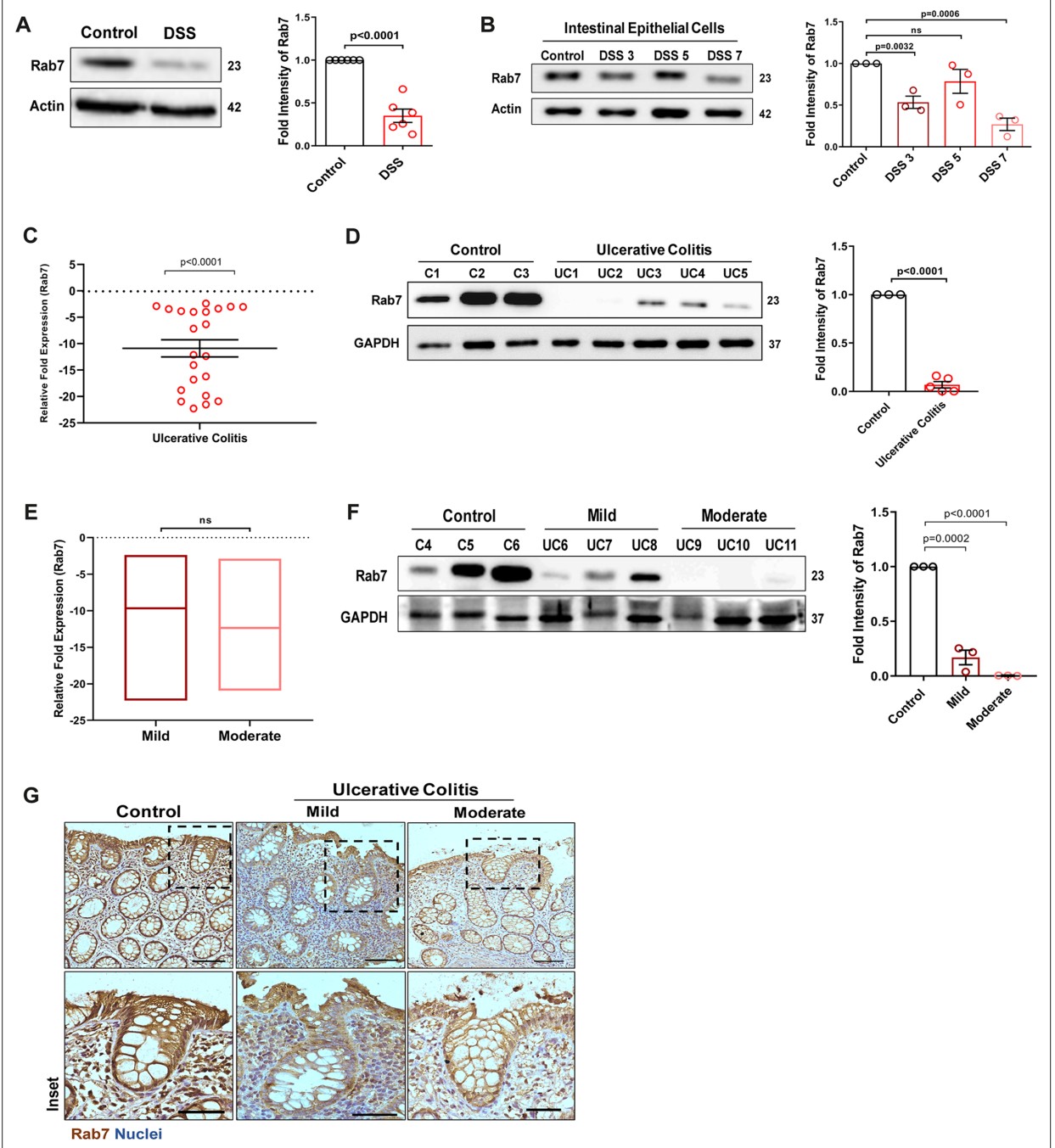

**Figure 1.** Small GTPase Rab7 shows altered expression during murine and human colitis correlative of disease severity. '(**A**) Rab7 expression analyzed in whole colon tissue of healthy and dextran sulfate sodium (DSS)-treated mice. Graph represents densitometric analysis showing fold intensity of *Rab7* expression calculated by normalizing to loading control (*β actin*). (**B**) Dynamics of Rab7 expression in intestinal epithelial cells isolated from the intestines of healthy and DSS-treated mice for different time durations. Corresponding graph shows densitometry analysis of Rab7 expression normalized to loading control (β actin). (**C**) RT-PCR analysis of relative fold expression of *Rab7* gene in human UC patient colonic biopsies (n = 22) relative to average control values (n = 22). *HPRT* was used for normalization. (**D**) Immunoblotting of Rab7 protein in human ulcerative colitis (UC) (n = 5) and control (n = 3) biopsy samples. GAPDH was used as loading control. Graph shows densitometry analysis of Rab7 expression relative to control. (**E**) RT-PCR analysis of *Rab7* gene expression variation with disease severity. (**F**) Immunoblot showing Rab7 protein expression dynamics of mild (n = 3) and moderate (n = 3) UC patient samples relative to controls (n = 3). Graph represents densitometric analysis showing fold intensity of Rab7 expression calculated by normalizing to loading control (GAPDH). (**G**) Immunohistochemistry images of colon biopsies from control (n = 3), UC mild (n = 3), and UC moderate (n = 3) patients stained for Rab7 (brown color) (Scale bar = 100 μm). Inset shows zoomed areas of the image (scale bar = 50 μm). Each dot represents (**A, B**) one mouse or (**C, D, and F**) one human. Error bars represent mean + SEM. Statistical analysis by Student's *t*-test. ns = nonsignificant.

*Figure 1 continued on next page*

*Figure 1 continued*

The online version of this article includes the following source data and figure supplement(s) for figure 1:

**Source data 1.** Original files for the western blot analysis in *Figure 1* (anti-Rab7, anti-actin and anti-GAPDH).

**Figure supplement 1.** Rab7 expression in various states of murine and human colitis.

**Figure supplement 1—source data 1.** Original files for the western blot analysis in *Figure 1—figure supplement 1* (anti-Rab7, anti-actin, and anti-GAPDH).

the intestinal epithelial cells (IECs) displayed significant reduction in expression of Rab7 at this stage (day 3) (*Figure 1B*).

To investigate if these observations are relevant to human IBD, we carried out a systematic analysis of Rab7 in human UC endoscopic samples. Details of the patient's clinical parameters are summarized in *Supplementary file 1*. A total of 65 human samples were utilized for different experiments. The control biopsies in our investigation were acquired from non-IBD patients. qRT-PCR of 22 colonoscopy tissue samples each of UC and controls revealed ~−11-fold downregulation of *Rab7* in colitis relative to healthy controls (*Figure 1C*). The downregulation was also evident at protein levels as revealed by immunoblotting (*Figure 1D*). In Crohn's disease (CD), another form of IBD, similar pattern was visible (*Figure 1—figure supplement 1G and H*).

A recent investigation has indicated elevated Rab7 levels in IBD patients, notably in the colon's crypt region (*Du et al., 2020*). This disparity in Rab7 expression could be due to the nature of the samples and the severity of tissue inflammation therein, as also suggested by our findings in DSS-mice dynamics model. To further validate, we evaluated any correlation between Rab7 expression and disease severity. UC patients were grouped into mild (score 2–4), moderate (score 5–6), and remission (score 0–1) based on ulcerative colitis endoscopic index of severity (UCEIS) score. At the transcriptional level, a decrease in *Rab7* expression was observed in mild UC patients, with even more reduction in moderate patients compared to controls (*Figure 1E*). The lowering of expression was correlative to disease severity as noticed by immunoblotting and immunohistochemistry (*Figure 1F and G*). Notably, similar expression alteration of Rab7 was observed in UC remission patients (*Figure 1—figure supplement 1I*). Taken together, our data suggests that variations in Rab7 expression levels can be attributed to the extent of gut inflammation.

## Goblet cells of intestine display dominant expression of Rab7

The results of the immunoblotting and qRT-PCR clearly indicate lowering of Rab7 protein levels during inflammation. Although this decrease could be a result of lower cellular expression, a decrease in the number of Rab7-positive cells or an increase in Rab7-negative cells. For a finer understanding, Rab7 was examined for its localization in specific cell types and defined regions of the intestine. Sections of small intestine, proximal, and distal colon part of healthy mice were immunostained for Rab7. In line with the earlier data, apart from intestinal crypts, IHC showed Rab7-specific staining in certain portions of the epithelial lining. A closer look revealed that the staining was higher in cells having a vacuolated morphology, a specific characteristic of goblet cells of the epithelium (*Figure 2—figure supplement 1A*). Further, colocalizing Rab7 with UEA1, a goblet cell-specific marker, confirmed dominant expression in secretory cells of intestine, that is, goblet cells (*Figure 2—figure supplement 1B*). Notably, Rab7 was seen in both crypt base and inter crypt goblet cells (*Figure 2—figure supplement 1B*, inset). We further examined for additional changes in sub-tissue level Rab7 protein expression pattern during colitis. Interestingly, immunofluorescence-based staining of Rab7 and goblet cells in colonic sections of DSS mice revealed significantly reduced levels of Rab7 distinctively in goblet cells compared to control mice (*Figure 2A and BFigure 2—figure supplement 1C*). Similar results were obtained in the case of UC patient colonic biopsy sections with respect to healthy controls (*Figure 2C and DFigure 2—figure supplement 1D*).

Goblet cells have a secretory role and as such involve a vigorous and tightly regulated intracellular vesicle trafficking system. Since Rab7 GTPase is a key regulator of cellular vesicle transport pathway (*Guerra and Bucci, 2016*), we investigated its possible role in goblet cell function in the context of intestinal inflammation. Pluripotent HT29 cells were cultured in media and condition that is known to induce their differentiation into goblet-like cells (*Phillips et al., 1988*). Post culturing, the cells showed vacuolated cell morphology along with a loss of expression of Lgr5, stem cell marker, and

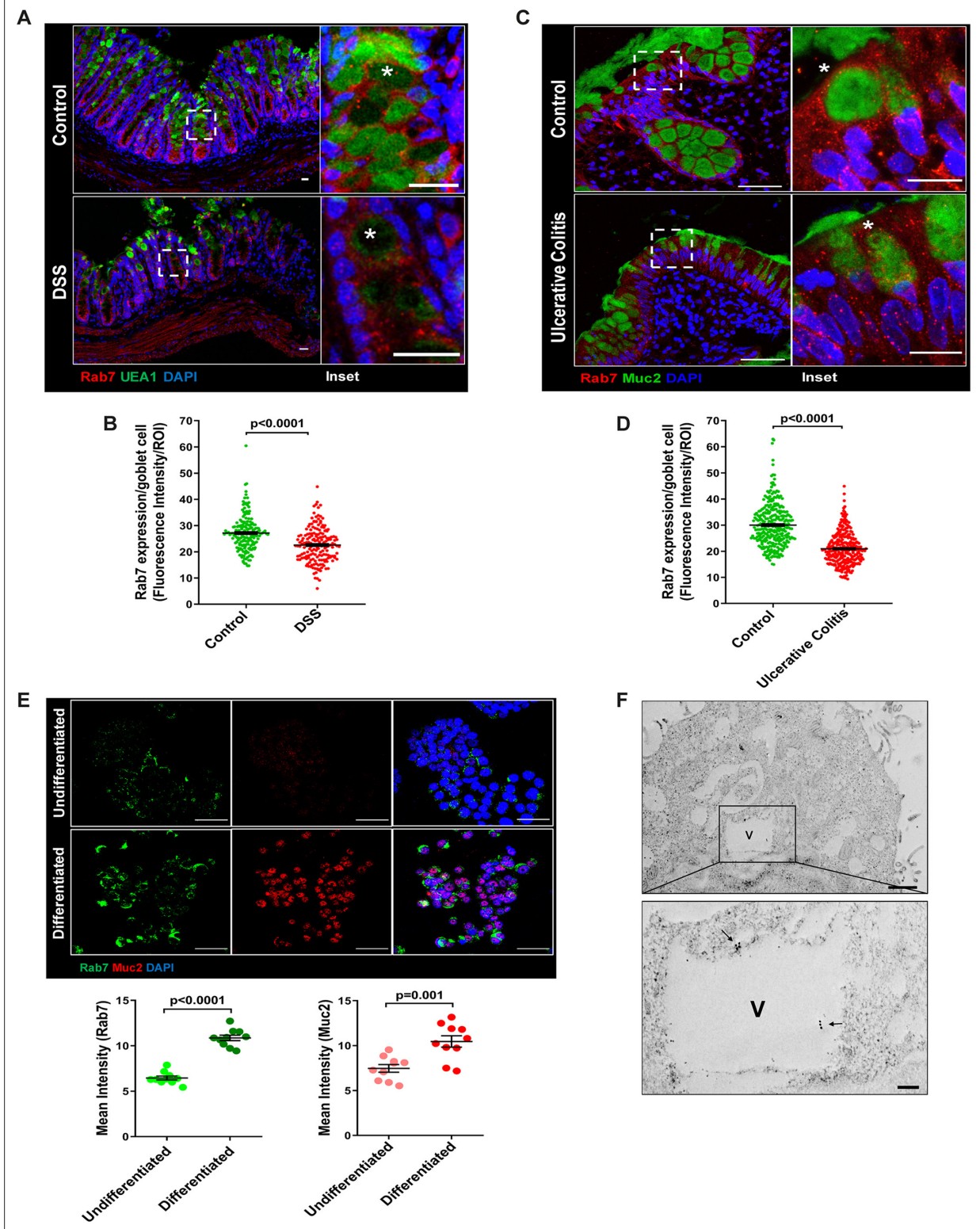

**Figure 2.** Goblet cells of intestine display dominant expression of Rab7. (**A, B**) Confocal imaging of healthy and dextran sulfate sodium (DSS)-treated mice (7 d) colon sections showing Rab7 (red) expression within goblet cells (UEA1-FITC) marked with asterisk. Inset shows zoomed areas of the image (scale bar = 20 μm). Graph shows Rab7 fluorescence intensity within UEA1 positive goblet cells measured within region of interest (ROI). 20 goblet cells were selected randomly from three different fields of each mouse (n = 3). (**C, D**) Representative co-immunofluorescence images of human ulcerative colitis (UC) and control colon biopsy sections stained for Rab7 (red) and Muc2 (green) for goblet cells (marked with asterisk) (scale bar = 50 μm). Inset

*Figure 2 continued on next page*

*Figure 2 continued*

shows zoomed areas of the image (scale bar = 20 μm). Graph shows Rab7 fluorescence intensity within Muc2-positive goblet cells measured within ROI. 20 goblet cells were selected randomly from five different fields of each sample (n = 3). (**E**) Co-immunofluorescence of Rab7 (green) and Muc2 (red) in undifferentiated and differentiated HT-29 cells (scale bar = 50 μm). Graphs show Rab7 and Muc2 fluorescence intensity measured in 10 different fields of three independent experiments each. (**F**) Rab7 protein visualized in HT29-MTX-E12 cells by immune-EM. Arrows indicate presence of Rab7 on vacuoles of cells (scale bar = 1 μm). Inset showed zoomed area of image (scale bar = 200 nm). Error bars represent mean + SEM. Statistical analysis by Student's *t*-test.

The online version of this article includes the following source data and figure supplement(s) for figure 2:

**Figure supplement 1.** Expression of Rab7 in goblet cells of different types and regions of gastrointestinal tract.

**Figure supplement 1—source data 1.** Original files for the western blot analysis in *Figure 2—figure supplement 1F* (anti-Rab7 and anti-actin).

a concomitant increase in Muc2, a goblet cell-specific marker (*Figure 2—figure supplement 1E*). Interestingly, compared to undifferentiated HT29 cells, these goblet-like cells showed an increased expression of Rab7 (*Figure 2E*, *Figure 2—figure supplement 1F*). To reconfirm these findings in an alternate cell type, we used HT29-MTX-E12 cells, a differentiated goblet-like cell line. In these cells, Rab7 was detectable using gold-labeled antibodies in immune-electron microscopy. Moreover, ultrastructural details (*Figure 2—figure supplement 1G*) showed Rab7 protein to be present in the vicinity of secretory vesicles (*Figure 2F*). Taken together, these data suggest that Rab7 may be involved in regulating the goblet cell function in the intestine.

## Knockdown of Rab7 in vivo aggravates DSS-induced colitis in mice

A reduction in Rab7 protein level during colitis, specifically in goblet cells, prompted us to further investigate its role in intestinal inflammation. Rab7 knockout mice are embryonic lethal; therefore, we developed a transient Rab7 knockdown mice model using an in-house nanogel based oral nucleic acid delivery system (*Kawamura et al., 2020*; *Yavvari et al., 2019*). Briefly, mice were fed with either scrambled (C^Scr) or *Rab7*-specific siRNA (Rab7^KD) engineered with nanogel. A sub-group of each category of mice was then fed with DSS (DSS+C^Scr, DSS+Rab7^KD) or left untreated (C). A schematic representation that summarizes the different treatments to mice is presented in *Figure 3A*. Interestingly, only 4 d of DSS administration was sufficient to generate severe inflammation in Rab7^KD mice which was atypical for DSS-colitis model. This was evident by reduced body weight (*Figure 3B*), diarrhea, and rectal bleeding (*Figure 3C*). These mice were dissected and relevant organs were harvested for further analysis. Intestinal epithelial cells were isolated from colon to check for the expression of Rab7 after siRNA treatment through western blot. Almost 50% of protein expression was reduced in Rab7^KD mice groups, thus confirming successful knockdown (*Figure 3F*). No significant alterations in Rab7 expression were visible in other organs such as spleen, MLN, and liver, proving no off-targeting of nucleic acid by nanogel (*Figure 3—figure supplement 1A*). DSS and DSS+C^Scr mice showed a significant reduction in colon length with inflamed appearance compared to the C, C^Scr, and Rab7^KD groups (*Figure 3D*). Notably, the features were more severe in the case of DSS+Rab7^KD group compared to DSS+C^Scr mice. In line with this, we observed splenomegaly in DSS+Rab7^KD group depicting higher inflammation (*Figure 3E*). H&E staining of distal colon of DSS+ Rab7^KD mice showed exacerbated signs of inflammation such as immune cell infiltration, colonic wall thickening, goblet cell and crypt loss, and epithelial erosion (*Figure 3G*). While many of these signs were evident in DSS+C^Scr mice, in the DSS+Rab7^KD group the intensity was higher as can be seen in the histopathology score plot (*Figure 3H*). Mucus scrapings from the colons of different groups of mice were examined for the presence of TNFα, the key proinflammatory cytokine. ELISA revealed a significant upregulation of TNFα in DSS+C^Scr mice, which was further higher in DSS+Rab7^KD mice (*Figure 3I*). These data led us to conclude that downregulation of Rab7 heightens DSS-induced colitis in mice, thus hinting toward its functional importance during colitis.

## Depletion of Rab7 in intestinal epithelium modulates mucus layer permeability and gut microbiota

Goblet cells primarily function to secrete mucins which release to form a mucus layer protective for the mucosal surface. Dysfunction of goblet cells leading to reduced mucus layer secretion and decreased barrier function is a frequent abnormality reported in UC (*Gersemann et al., 2009*). We observed

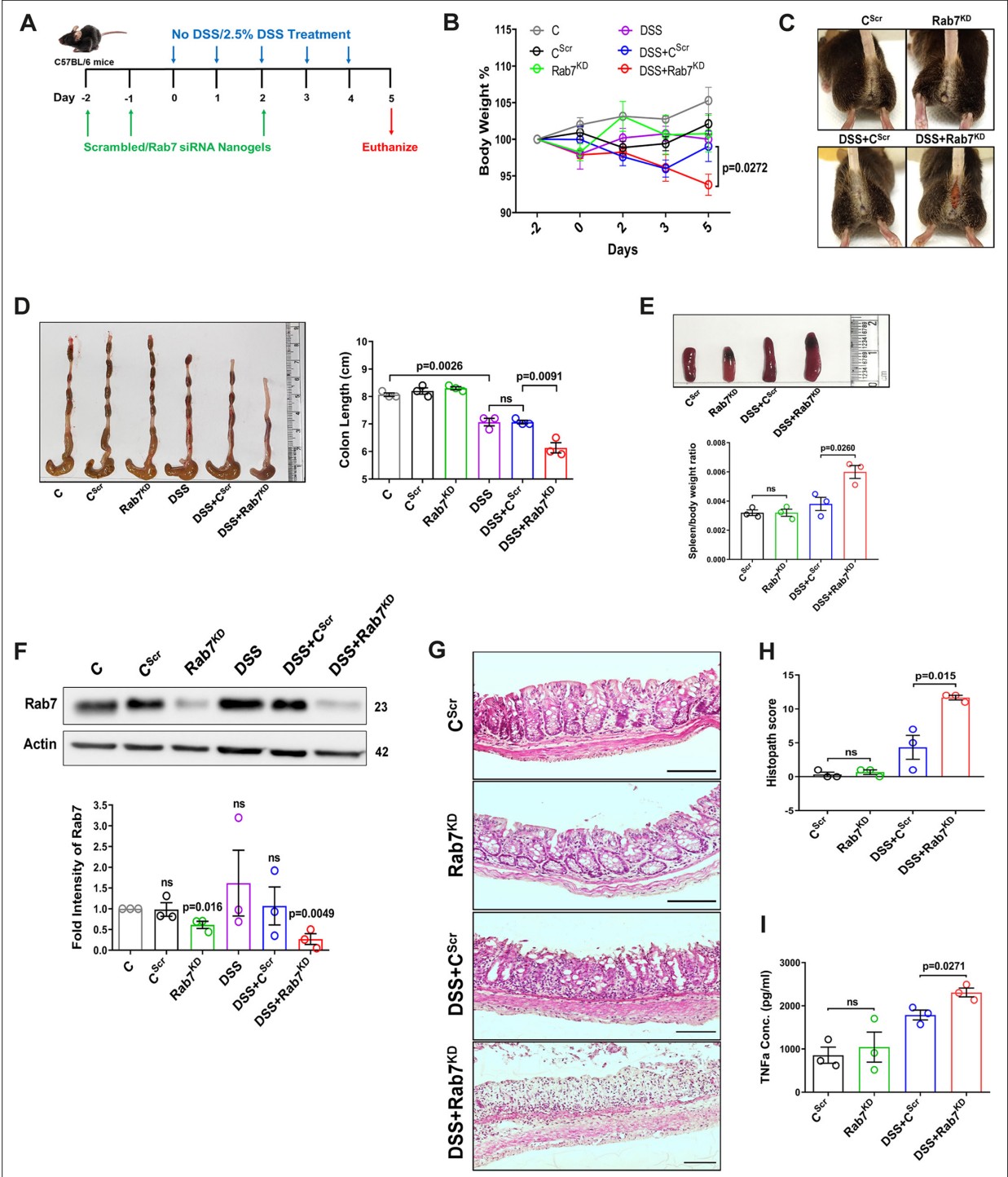

**Figure 3.** Downregulation of Rab7 aggravates inflammation upon external trigger. (**A**) Schematic representation of the experimental plan for knockdown of Rab7 in C57Bl/6 mice showing different treatments (n = 3 mice per group). (**B**) Graph showing body weight percent. (**C**) Representative photographs demonstrating rectal bleeding in mice. (**D**) Gross morphology of colon and caeca. Graph shows colon length quantification. (**E**) Representative spleens of different treatment mice groups. Graph showing spleen to body weight ratio. (**F**) Rab7 protein expression in isolated intestinal epithelial cells from mice colon. Graph represents densitometric analysis showing fold intensity of Rab7 expression calculated by normalizing to loading control (β actin). Significance value of each group is relative to the untreated (C) group. (**G, H**) Hematoxylin and eosin staining of distal colon sections with histopathology scores showing increased characteristics of inflammation. (**I**) ELISA of TNFα from mucosal extracts of mice colon. Each dot represents one mouse. Error bars represent mean + SEM. Statistical analysis by (**B**) two-way ANOVA or Student's t-test. ns = nonsignificant.

The online version of this article includes the following source data and figure supplement(s) for figure 3:

*Figure 3 continued on next page*

*Figure 3 continued*

**Source data 1.** Original files for the western blot analysis in *Figure 3F* (anti-Rab7 and anti-actin).

**Figure supplement 1.** Rab7 knockdown in mice is specific to intestine.

**Figure supplement 1—source data 1.** Original files for the western blot analysis in *Figure 3- figure supplement 1A* (anti-Rab7 and anti-actin).

a conspicuously higher expression of Rab7 in goblet cells in steady state, while reduction during colitis as seen in *Figure 2B and D*. Based on these data, we hypothesized that Rab7 perturbation would impact goblet cell function and mucus secretion. To test this, histological examination of the colon from various groups described above was carried out through Alcian-blue staining for mucins. DSS-treated mice groups showed a significantly decreased staining, indicating reduced goblet cell number. Notably, this phenotype was observed in Rab7[KD] mice colon also, although the data was not significant (*Figure 4—figure supplement 1A and B*). For the analysis of the mucus, thickness of the inner mucus layer (IML) was measured in Alcian blue-stained colon specimens. As expected, the decrease in goblet cell number was accompanied by thinning of the mucus layer in both the DSS-treated mice groups (*Figure 4A*). Surprisingly, in Rab7[KD] mice IML was significantly much wider and seemed to be less dense showing a lighter stain of Alcian blue (*Figure 4B*). This led us to expect alterations in the permeability of the mucus layer. We speculated that depletion of or lack of Rab7 may be

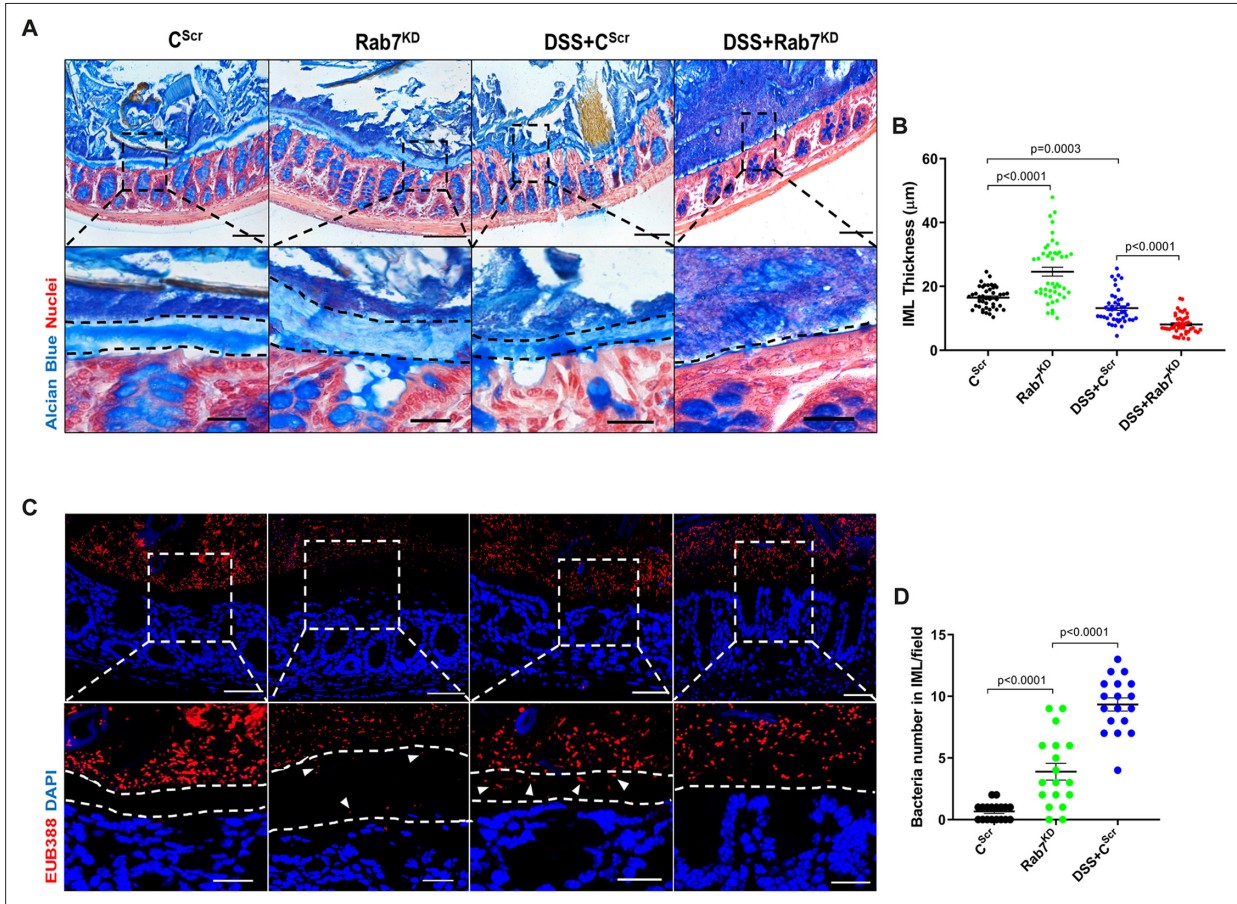

**Figure 4.** Rab7 downregulation in intestinal epithelium modulates mucus layer thickness and permeability. (**A–B**) Alcian blue staining in mice distal colon sections displaying inner mucus layer (IML) (blue) marked with dotted lines and nuclei (red). Thickness of IML was measured using ImageJ software (15 measurements per section of each mice). Scale bar = 100 μm and 20 μm (inset). (**C, D**) Representative images of fluorescence in situ hybridization (FISH) staining for bacteria detection in mucus layer using general bacterial probe EUB338-Alexa Fluor 647 (red). Arrowheads demarcate the presence of bacteria in IML (dotted lines). Graph shows bacteria count detected in IML (six regions per section of each mouse). Scale bar = 100 μm and 20 μm (inset). Error bars represent mean + SEM. Statistical analysis by Student's *t*-test.

The online version of this article includes the following figure supplement(s) for figure 4:

**Figure supplement 1.** Rab7 perturbation changes goblet cell number and Muc2 distribution.

resulting in compromised mucus properties. To test this, initially a Rab7 knockout HEK cell line (hereafter HEK RAB7$^{-/-}$) was generated. Knockout in intestinal epithelial cells was attempted but we were unable to obtain a line without Rab7 expression. HEK RAB7$^{-/-}$or HEK WT cells were transfected with pSNMUC2-MG plasmid (a kind gift from Professor Gunnar C Hansson, University of Gothenburg). This plasmid expresses human Muc2 N-terminal domains fused with GFP (*Godl et al., 2002*). Surprisingly, the Muc2 protein was seen to be not forming a theca and was scattered over cell cytoplasm compared to wildtype cells corroborating the above in vivo data (*Figure 4—figure supplement 1D*).

To check for any alterations in the permeability of the IML of Rab7$^{KD}$ mice, we analyzed the colon sections, using published protocol, for bacterial presence in the mucus layer by fluorescence in situ hybridization (FISH) involving a universal bacterial probe EUB388. In control mice as anticipated, bacteria were detected in the outer mucus layer but not in inner layer (*Figure 4C*). On the contrary, in Rab7$^{KD}$ mice, in spite of the mucus layer being wider several bacteria were detected in the inner mucus layer verifying increased permeability (*Figure 4D*). DSS+C$^{Scr}$ mice showed a dramatic increase in the number of bacteria penetrating the inner layer. As there was complete invasion of bacteria to epithelium surface in case of DSS+Rab7$^{KD}$-treated mice, quantification of the number of penetrating bacteria was unworkable. These data clearly indicated toward the involvement of Rab7 in modulating the mucus-secreting function of goblet cells which further alters the mucus layer characteristics leading to bacterial breach and generation of inflammation.

Besides mucus barrier integrity, intestinal microbiota plays an important role in gut inflammation (*Al Bander et al., 2020*). We analyzed the microbial community in stool samples of Rab7$^{KD}$ and DSS+C$^{Scr}$ compared to C$^{Scr}$ mice. Our main motive here was to specifically examine alterations in microbial diversity/abundance in Rab7$^{KD}$ mice relative to DSS+C$^{Scr}$ and C$^{Scr}$ mice. Fecal samples were collected at day 5 of DSS treatment in the Rab7$^{KD}$ mice model, particularly since these mice display significant inflammation by day 5 due to Rab7 deficiency. As anticipated, Shannon index plot showed decreased microbial diversity richness and evenness in the DSS+C$^{Scr}$ mice groups ($\mu_{median}$ = 3.67). This phenotype was more severe in Rab7$^{KD}$ mice ($\mu_{median}$ = 3.46) (*Figure 5A*). Further, principal coordinates analysis (PCoA) indicated that each of the three groups assumed a discrete cluster (*Figure 5B*). Similar divergence among DSS+C$^{Scr}$ and C$^{Scr}$ were seen in non-metric multidimensional scaling (NMDS) plots (*Figure 5—figure supplement 1A*). While each group displayed the presence of some unique operational taxonomic units (OTUs), some shared OTUs were also notable (*Figure 5C*). Beta diversity index heatmap revealed differences in the microbial diversity among different experimental mice groups (*Figure 5—figure supplement 1B*). The map clearly indicated a shift in microbial diversity in Rab7$^{KD}$ mice such that they appeared to occupy an intermediate position between C$^{Scr}$ and DSS+C$^{Scr}$ mice. Substantial differences in the relative abundance of many microbes were found (*Figure 5D*). At the phylum level, bacteroidetes showed an increase in abundance, whereas firmicutes were significantly less represented in the Rab7$^{KD}$ and DSS+C$^{Scr}$ mice groups (*Figure 5E*). The other analyzed phyla did not show any significant changes. Firmicutes are known to get less abundant in IBD (*Frank et al., 2007*, *Manichanh et al., 2006*). Class level analysis revealed that the firmicutes reduced were clostridia, while the abundance of Bacteroidia was more among Bacteroidetes (*Figure 5F*). Abundance of several other microbes getting altered could be appreciated at order, family, genus, and species level (*Figure 5—figure supplement 1C*). Genus such as *Lactobacillus* and *Parabacteroidetes* were found to be significantly altered in Rab7$^{KD}$ mice compared to C$^{Scr}$ (*Figure 5G*). Intriguingly, compared to C$^{Scr}$ mice, Rab7$^{KD}$ and DSS+C$^{Scr}$ mice showed a decrease in the abundance of a particular category of bacterial species (*Figure 5—figure supplement 1D*, red box). The majority of these contained probiotic microorganisms. These include *Bifidobacterium longum* and *Lactobacillus helveticus*, both of which have been thoroughly researched in relation to colitis (*Ho et al., 2022*; *Tamaki et al., 2016*; *Zhang et al., 2021*). All these data demonstrated aberrant composition of gut microbiota of Rab7$^{KD}$ mice with more similarity with DSS mice inflamed gut microbiota.

## Rab7 perturbation impacts mucus composition in colon

We next sought to investigate in detail the alterations in the colonic mucus composition in Rab7 perturbed mice which might influence penetrability and gut microbiota. The secreted mucus is composed of a mixture of proteins, including those contributing to mucus gel architecture, antimicrobial peptides, and regulatory processes. We went on to investigate any modifications in the total amount or composition of the secreted mucus of Rab7$^{KD}$ mice. Mice colon was harvested and secreted

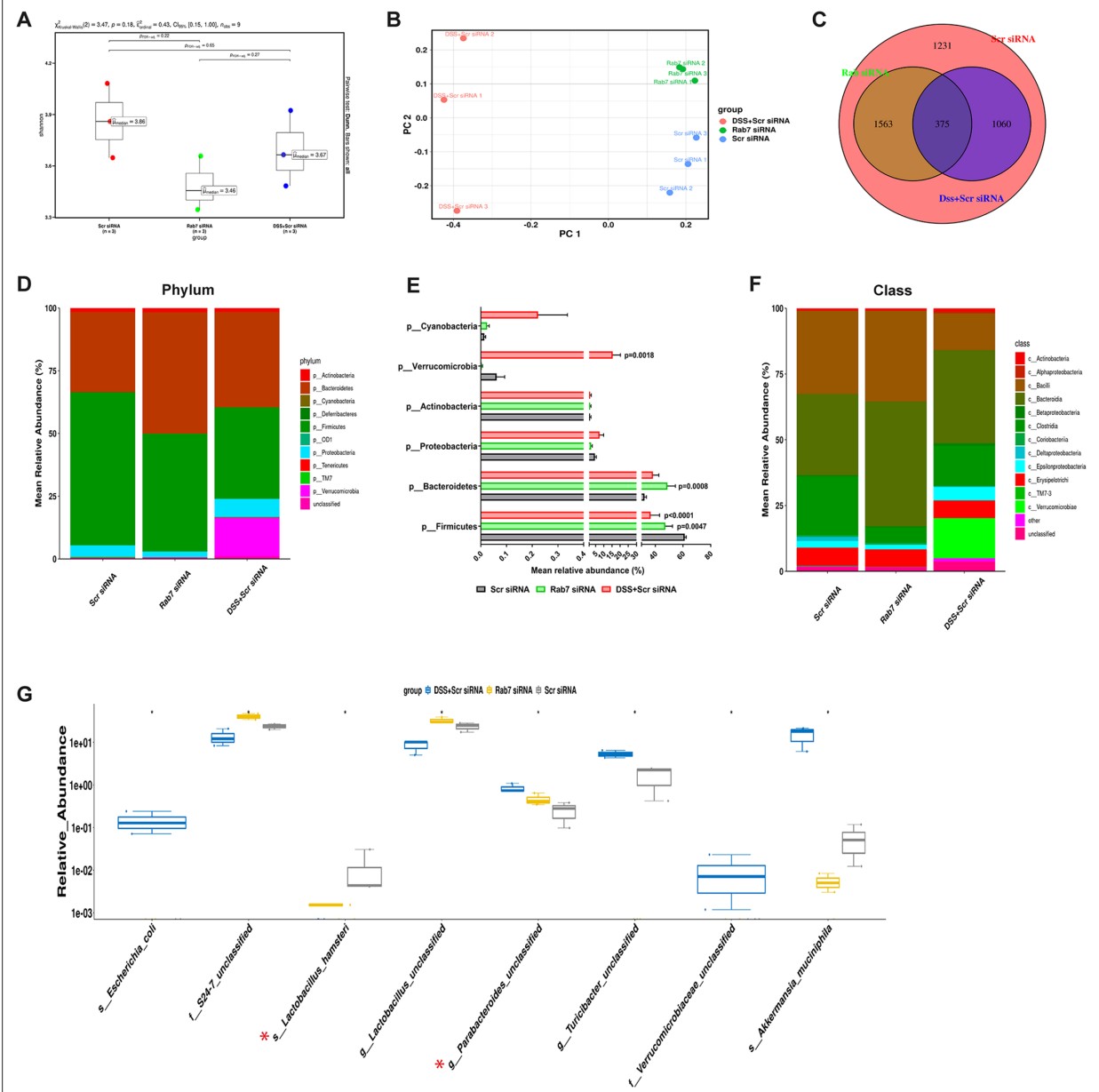

**Figure 5.** Gut microbiota in Rab7 knockdown mice is altered alike dextran sulfate sodium (DSS)-colitis mice. Microbial composition of C^Scr (Scr siRNA), Rab7^KD (*Rab7* siRNA), and DSS+C^Scr (DSS+Scr siRNA) was analyzed by 16S metagenomic profiling. (**A**) Alpha diversity quantified as Shannon index. Significance was calculated using the Kruskal–Wallis test followed by the improved Benjamini–Hochberg procedure from false discovery rate (FDR) correction. (**B**) Principal coordinates (PCoA) plot calculated from distance matrices obtained from Bray–Curtis. (**C**) Venn diagram representation of shared and unique operational taxonomic units (OTUs) between groups. (**D–F**) Mean relative abundance of each taxon. Relative abundance of top 10 phylum (**D**) and class (**F**) of each group is depicted in the figure. Each bar represents the mean of the merged OTUs from three mice. Taxonomic lineages not included in top 10 were collapsed as 'Others', while the ones which have not been classified has been placed under the category of 'unclassified'. Relative abundance (in %) of some important taxa showing significant differences between experimental groups (**E**). (**G**) Plot showing relative abundance of eight different taxa, identified with Kruskal–Wallis test. Each dot represents one mouse.

The online version of this article includes the following source data and figure supplement(s) for figure 5:

**Source data 1.** Relative abundance of all bacterial identities (numerical source data).

**Figure supplement 1.** Gut microbiota composition analysis of Rab7KD mice using 16S rRNA sequencing.

mucus was isolated by gentle scrapping using a rubber policeman (*Figure 6A*) and run on a mucin gel stained with Alcian blue dye. No significant change was observed in the total mucin content between any of the groups (*Figure 6B*). We speculated that there may be changes in specific components of mucus. For identification of proteins in the secreted mucus, isolated mucus samples were subjected to reducing buffer and separated on SDS-PAGE followed by high-resolution tandem mass spectrometry (mucus proteomics) (*Figure 6C*). Raw data was analyzed using MaxQuant software (v1.6.0.16). Masses were searched against the mouse UniProt proteome database and additional databases such as miceMucinDB and VerSeDa. A total of 522 proteins were detected among different mice groups (*Figure 6D*), of which 288 were common in all along with some unique proteins in each group as is evident through the Venn diagram (*Figure 6E*). Principal component analysis (PCA) showed proper grouping of the biological replicates from different experimental groups (*Figure 6—figure supplement 1A*). Label-free quantification identified approximately 500 differentially expressed proteins among the samples (*Figure 6F*). Significantly altered proteins were analyzed with false discovery rate (FDR) (*Figure 6G*). Proteins observed to be differentially expressed in Rab7$^{KD}$ mice compared to C$^{Scr}$ mice or DSS+Rab7$^{KD}$ mice and DSS+C$^{Scr}$ mice are represented in respective volcano plots (*Figure 6H and I*). These proteins were part of certain pathways over-represented in Gene Ontology (GO) analysis like extracellular exosome biogenesis, vesicle localization, and calcium-dependent phospholipase A2 activity based on biological processes (*Figure 6—figure supplement 1B*) and molecular function (*Figure 6—figure supplement 1C*). All the proteins which displayed significant differential expression were localized in different subcellular compartments (*Figure 6—figure supplement 1D and E*). Some cytoplasmic and nuclear proteins were also represented in this, which may be debris or exudates from lysed cells. Certain secretory proteins such as chloride channel accessory 1 (CLCA1), apolipoprotein AI (APOA1), and transferrin (Tf) caught our interest as they were seen to be upregulated in the Rab7$^{KD}$ mice group when compared with the C$^{Scr}$ (highlighted in red in *Figure 6—figure supplement 1D and E*). As, transferrin or Tf is an iron-binding glycoprotein secreted from the liver, their increased amount in mucus indirectly hinted towards the decrease in transferrin receptors (TFRC) in the intestinal cells which uptake them. We therefore examined the expression level of *CLCA1*, *APOA1,* and *TFRC* in colon tissue through qRT-PCR analysis. While *TFRC* showed down-expression in DSS+Rab7$^{KD}$ mice, no significant change was observed for *CLCA1* and *APOA1* (*Figure 6—figure supplement 1F–H*). In case of *CLCA1*, several folds downregulation at RNA level in Rab7$^{KD}$ and DSS+Rab7$^{KD}$ mice was seen, although the changes were nonsignificant. CLCA1 is a protease exclusively secreted by goblet cells and forms one of the major mucus components. It has recently been demonstrated that CLCA1 helps in maintaining the dynamics of mucus layer in colon by processing Muc2. The CAT/Cys and VWA domain containing part of CLCA1 protein (53 kDa) has maximum Muc2 cleaving property. We went on to check for the expression of CLCA1 at protein levels in the mucus of Rab7$^{KD}$ mice. Remarkably, CLCA1 (53 kDa) was found to be highly upregulated in mucus samples of Rab7$^{KD}$ followed by DSS-treated mice compared to controls (*Figure 7A*). CLCA1 could thus be an interesting and probable candidate to justify modifications in mucus layer with its increased secretion in mucus upon Rab7 knockdown in mice.

## Altered Rab7 influences CLCA1 expression in goblet cells

We investigated the expression alteration of CLCA1 in human UC. Earlier, *van der Post et al., 2019* reported CLCA1 to be getting downregulated in active UC patients. In line with this, qRT-PCR analysis of *CLCA1* in 22 UC and control biopsy samples each revealed significant downregulation in UC relative to healthy controls (*Figure 7B*). Next, expression of CLCA1 protein was investigated in tissue and secreted mucus from UC patients (same as used for analyzing Rab7 expression in *Figure 1D*). To achieve that, human colon biopsies were treated with N-acetyl cysteine, a mucolytic agent, to isolate secreted mucus (*Figure 7—figure supplement 1A*). The mucus samples obtained this way and the remaining tissues were processed separately for immunoblotting. Interestingly, similar to the Rab7 knockdown mice, a higher expression of CLCA1 was seen in these human UC samples compared to healthy controls both in tissue and secreted mucus (*Figure 7C*). Moreover, there was an inverse correlation between expressions of Rab7 and CLCA1 both in tissue and secreted mucus of human colon biopsies (*Figure 7D*). Also, the immunostaining of Rab7 and CLCA1 in human UC and control colon sections revealed higher expression of CLCA1 in crypts and goblet cells of UC compared to healthy controls, opposite to the pattern seen for Rab7 (*Figure 7E*). These data

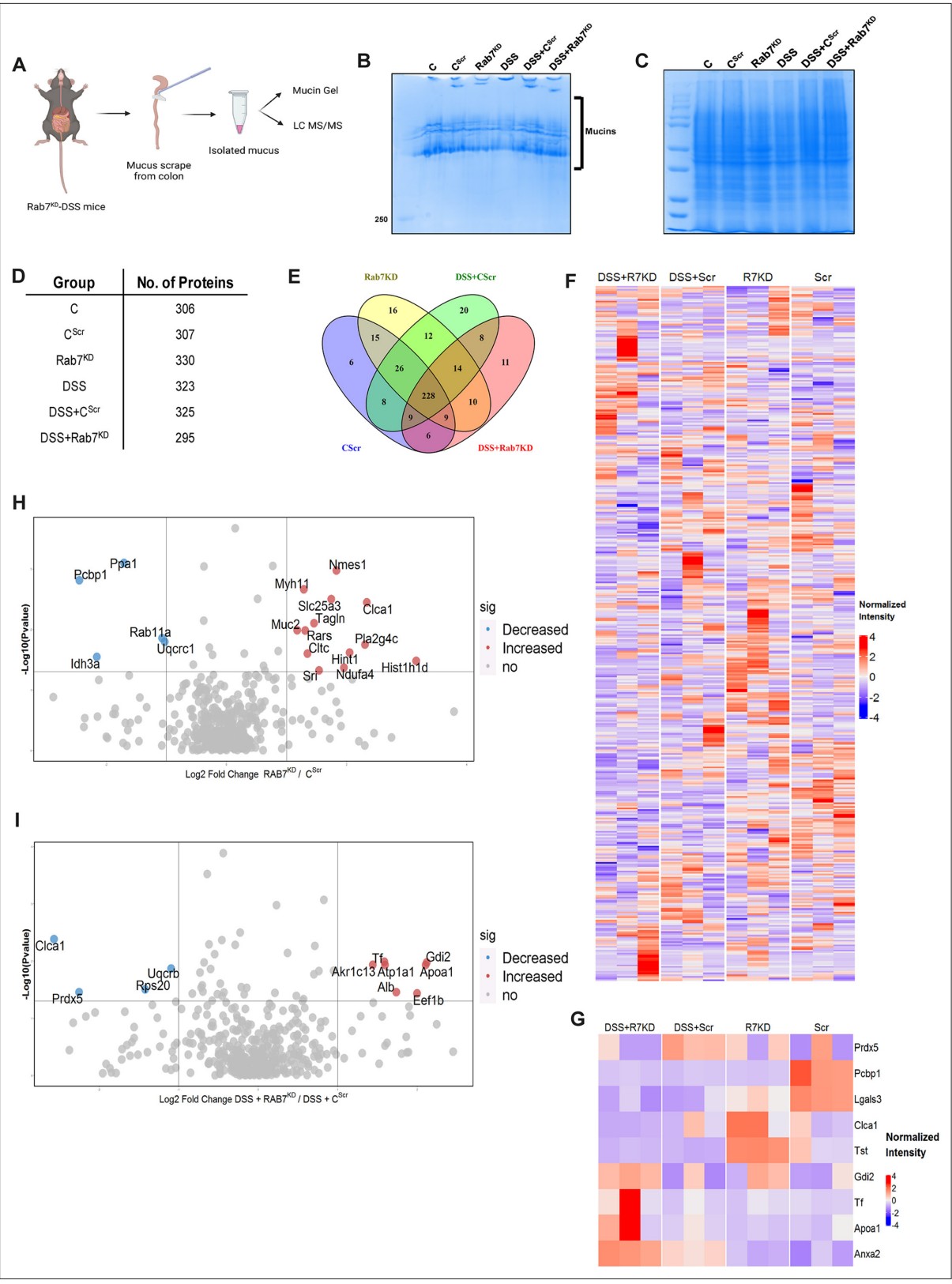

**Figure 6.** Rab7 perturbation alters mucus composition of colon. (**A**) Schematic representing steps for mucus isolation from mice colon followed by sample preparation for total mucin measurement through mucin gels and mucus layer composition analysis using mass spectrometry (Created with BioRender.com and published using a CC BY-NC-ND license with permission). (**B**) Mucin gel showing the amount of total mucins (blue) stained with Alcian blue dye. (**C–E**) Coomassie staining of mucus samples isolated from different experimental groups. Table listing the total number of proteins

*Figure 6 continued on next page*

*Figure 6 continued*

identified in the mucus samples through mass spectrometry and Venn diagram displaying the number of individual proteins common and unique in different samples. (**F, G**) Heat maps of the proteins differentially expressed without false discovery rate (FDR) (**F**) and with FDR (**G**). (**H, I**) Volcano plots of proteome in mucus samples of different mice groups showing differential expression of proteins in Rab7KD mice versus CScr (**H**) and DSS+Rab7KD mice versus DSS control group (**I**).

The online version of this article includes the following source data and figure supplement(s) for figure 6:

**Source data 1.** List of all the proteins identified in different samples of mucus (numerical source data).

**Figure supplement 1.** Mucus proteome analysis of Rab7KD mice.

depicted that CLCA1 protein expression is increased in tissue, resulting in higher secretion into mucus during colitis.

To examine the mechanism of possible Rab7-mediated control of CLCA1 secretion, we used HT29-MTX-E12, cultured goblet cells. siRNA-mediated knockdown of Rab7 was carried out in these cells, followed by analysis of CLCA1 protein expression. We observed an increase in CLCA1 expression in the Rab7 knockdown cells (*Figure 7F*). Further, overexpression of Rab7-GTP locked form (active Rab7) lowered CLCA1 in cells (*Figure 7G*). All this data led us to anticipate the involvement of Rab7 in regulating the CLCA1 expression in the goblet cells of intestine.

Since currently there are no reports addressing CLCA1 protein stability in goblet cells, we carried out experiments to delineate this further. HT29-MTX-E12 cells were treated with MG132 (proteasome inhibitor) and bafilomycin (lysosome inhibitor) and CLCA1 expression was checked. CLCA1 showed rescue upon bafilomycin treatment, confirming that it gets degraded via lysosomal pathway (*Figure 7H*). No significant change was seen in CLCA1 expression upon proteasomal pathway inhibition (*Figure 7—figure supplement 1C*). We speculated a role for Rab7 in the regulation of CLCA1 protein turnover via lysosomal degradation pathway since Rab7 GTPase is known to facilitate fusion of late endosomes to lysosomes for degradation. Structured illumination microscopy (SIM) images revealed colocalization of Rab7-CLCA1 with lysosomes (*Figure 7I*). Further, Rab7 was knocked down in HT29-MTX-E12 cells and the process of CLCA1 fusion with lysosomes was imaged using confocal microscopy. The images revealed less colocalization of CLCA1 with lysosomes upon Rab7 knockdown compared to vehicle control (*Figure 7J and K*). Together, these data suggested modulation of CLCA1 protein degradation by Rab7 in goblet cells. Overall, our data highlights a crucial role of Rab7 in maintaining gut homeostasis. Rab7 downregulation, as observed during colitis, results in increased CLCA1 in goblet cell and thereby a higher secretion. These changes adversely affect the mucus layer, the composition of microbiota and epithelial barrier function altogether leading to inflammation (*Figure 8*).

## Discussion

Here we report the significant contribution of one of the essential cellular proteins, that is, Rab7 GTPase in colitis pathogenesis. Rab GTPases are amongst the fundamental proteins of a cell regulating important cellular mechanisms like membrane trafficking (*Hutagalung and Novick, 2011*). A number of comprehensive studies summarizing the molecular, physiological, and pathological aspects of vesicle transport system link trafficking proteins dysfunction with human diseases such as carcinoma, neurodegenerative disorders, Charcot–Marie–Tooth type B and diabetes (*BasuRay et al., 2013*; *Mafakheri et al., 2018*; *Millecamps and Julien, 2013*; *Tzeng and Wang, 2016*). Interestingly, the involvement of Rab GTPases in the underlying mechanisms implicating IBD is still not well understood. Rab5, Rab34, and Rab11 have earlier been reported to contribute to correct assembly and maintenance of junctional complexes in intestinal epithelial cells (*Citalán-Madrid et al., 2013*, *Talmon et al., 2012*). Dislocation of Rab13 has been shown in patients with CD (*Ohira et al., 2009*).

Rab7 protein was strikingly observed to be downexpressed during human and murine colitis. To explain the discordance about Rab7 getting upregulated in crypts of IBD patients as observed by *Du et al., 2020*, we systematically analyzed different phases of inflammation development at diverse regions of intestinal mucosa. It seems Rab7 downregulation is dependent on stages of inflammation progression and the pattern varies between regions of the intestine viz. epithelium and crypts.

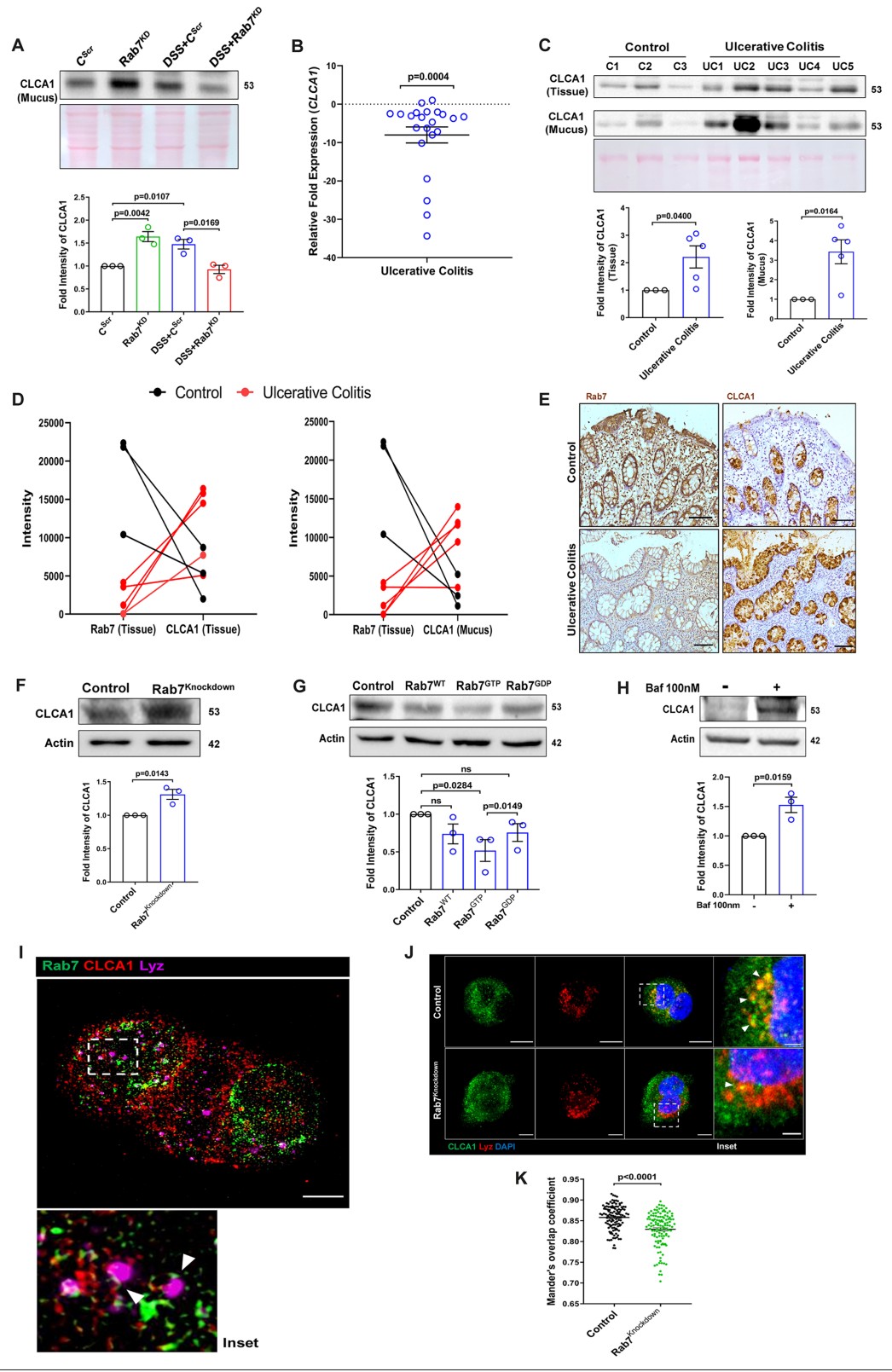

**Figure 7.** Rab7 mediates CLCA1 degradation via lysosomal pathway in goblet cells. (**A**) CLCA1 expression in mucus samples of Rab7^KD-DSS mice. Graph represents densitometric analysis showing fold intensity of CLCA1 expression. (**B**) RT-PCR analysis of relative fold expression of *CLCA1* gene in human ulcerative colitis (UC) (n = 22) patient colonic biopsies relative to average control (n = 22) values. *HPRT* was used for normalization.

*Figure 7 continued on next page*

*Figure 7 continued*

(**C**) Immunoblotting of CLCA1 protein in human UC (n = 5) and control (n = 3) mucus and biopsy samples. Graphs represent densitometric analysis showing fold intensity of CLCA1 expression in mucus and tissue samples calculated by normalizing to loading control (GAPDH shown in *Figure 1F*). (**D**) Correlation graph showing expression of Rab7 and CLCA1 in mucus and tissue samples of human UC patients relative of controls plotted using fold change from immunoblots. (**E**) Representative immunohistochemistry images of Rab7 and CLCA1 staining (brown color) in human UC patient (n = 3) and control biopsy (n = 3) sections and cell nuclei (blue color) (scale bar = 100 µm). (**F**) CLCA1 protein expression in HT29-MTX-E12 cells transfected with either scrambled siRNA (control) or Rab7 siRNA (Rab7$^{Knockdown}$). Graph represents densitometric analysis showing fold intensity of CLCA1 expression calculated by normalizing to loading control (β actin). (**G**) Immunoblot showing CLCA1 protein expression change in HT29-MTX-E12 cells overexpressed with EGFP empty vector (control), Rab7-GFP (Rab7$^{WT}$), Rab7-GFP GTP locked form (Rab7$^{GTP}$), and Rab7-GFP GDP locked form (Rab7$^{GDP}$). Graph represents densitometric analysis showing fold intensity of CLCA1 expression calculated by normalizing to loading control (β actin). (**H**) Immunoblot showing CLCA1 protein expression after treatment of bafilomycin in HT29-MTX-E12 cells. Graph represents densitometric analysis showing fold intensity of CLCA1 expression calculated by normalizing to loading control (β actin). (**I**) Representative image of structured illumination microscopy showing images of HT29-MTX-E12 cells transfected with Rab7$^{GTP}$ (green) and stained with CLCA1 (red) using anti-CLCA1 antibody and lysosomes with LysoTracker Red DND-99 (magenta) (scale bar = 5 µm). Inset shows zoomed areas of colocalization marked with arrows. (**J, K**) Representative confocal images of HT29-MTX-E12 cells transfected with either scrambled siRNA (control) or *Rab7* siRNA (Rab7$^{Knockdown}$). Cells are stained with CLCA1 (green) using anti-CLCA1 antibody and lysosomes with LysoTracker Red DND-99 (red). Graph shows quantitation of colocalization between CLCA1 and lysosomes from images (n = 120) using Mander's overlap coefficient (scale bar = 100 µm). Inset shows zoomed areas of the image with colocalization puncta (yellow) marked with arrows (scale bar = 50 µm). Each dot represents (**A**) one mouse or (**B, C, E**) one human or (**F–H**) one independent experiment. Error bars represent mean + SEM. Statistical analysis by (**C**) Welch's *t*-test or (**A, B, F–H, K**) Student's *t*-test. ns = nonsignificant.

The online version of this article includes the following source data and figure supplement(s) for figure 7:

**Source data 1.** Original files for the western blot analysis in *Figure 7* (anti-CLCA1 and 1023 anti-actin).

**Figure supplement 1.** CLCA1 protein degrades via lysosomal degradation pathway.

**Figure supplement 1—source data 1.** Original files for the western blot analysis in *Figure 7—figure supplement 1B* (anti-CLCA1 and anti-actin).

---

Using an inhibitor (CID1067700) of Rab7, its role in impairing B-cell class switching during murine lupus has been reported (*Lam et al., 2016*). Though specific to inhibit Rab7 GTPase activity, using this inhibitor limits the user to distinctively target any specific organ. In the current study, we have exploited a recently created technique to deliver nucleic acids to intestine. Although being transient, this mouse model appears promising to briefly study and understand the outcomes of gut-specific knockdown of Rab7. Successful knockdown of Rab7 in mice intestine revealed several cues regarding its role in intestinal homeostasis. DSS treatment to Rab7$^{KD}$ mice displayed worsening of inflammation in the intestine. This explains that downexpression of Rab7 during colitis can be the cause of disease.

UC pathogenesis is majorly associated with decreased number of goblet cells, which in turn attributes to compromised mucus layer. A recent report describes that the reduction in mucus layer is related to reduced secretory function of remaining goblet cells (*Birchenough et al., 2015*). Though ubiquitously expressed in all cell types, dominant expression of Rab7 in goblet cells of intestine was unpredicted. The fact that Rab7 was specifically getting altered in goblet cells during colitis was rather intriguing and necessitated us to investigate in this aspect. Our findings revealed alterations in thickness and permeability, leading to disrupted sterility in the inner mucus layer of Rab7$^{KD}$ mice. Further, dysbiosis in luminal microbiota was also evident through 16S metagenomics analysis. Reduction in firmicutes is the most consistent observation in IBD patients. Our data shows significantly decreased abundance of Clostridia, a class of Firmicutes in Rab7$^{KD}$ mice.

Alterations in the mucus layer characteristics could be attributed to changes in its composition. A report on the alterations of colonic mucus composition in UC shows that the structural weakening of mucus layer is due to reduction in 9 proteins among the 29 mucus core proteins such as Muc2, FCGBP, ZG16, and CLCA1. Mass spectrometry analysis revealed changes in the mucus composition upon Rab7 perturbation in mice colon. It was seen that CLCA1, among the few altered proteins, was surprisingly upregulated in Rab7$^{KD}$ mice. Several omics studies in human UC patients have suggested variations in the CLCA1 protein expression during inflammation (*Massimino et al., 2021*). In an attempt to

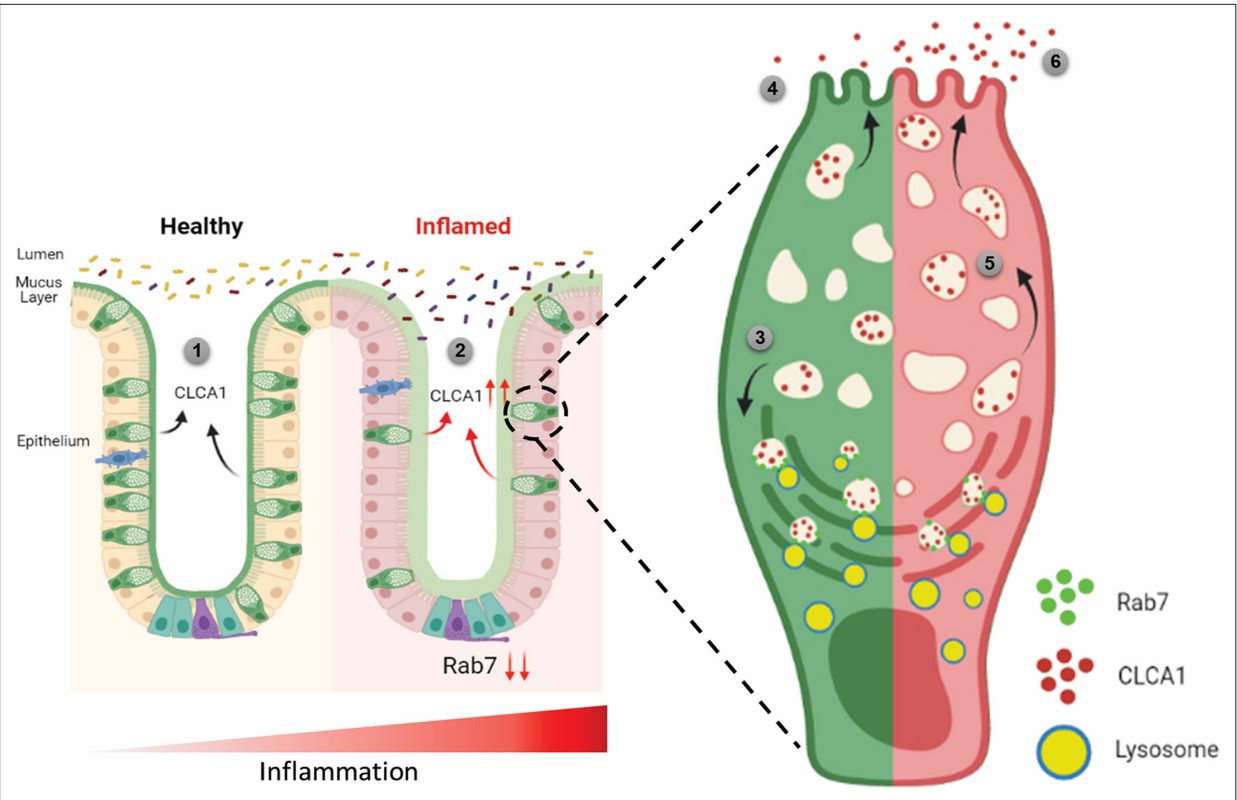

**Figure 8.** Rab7 maintains mucus layer dynamics in intestine by regulating degradation of CLCA1 protein via lysosomal fusion. Healthy intestine inhabits lumen microbes well separated by mucus layer secreted by goblet cells along with CLCA1 protein in balanced levels (1). During colitis, Rab7 downregulates along with increased expression of CLCA1, resulting in diffused mucus layer penetrable to microbes (2). In a goblet cell, CLCA1-filled vacuoles destined for secretion are rerouted for degradation pathway by Rab7 and fuse with lysosomes (3), leading to a balanced release outside the cell (4). However, during inflammation the loss of Rab7 consequently impedes CLCA1 degradation (5), fostering its increased secretion from the cell (6) (created with BioRender.com and published using a CC BY-NC-ND license with permission).

investigate this protein in colitis, a series of studies negated the involvement of CLCA1 in gut inflammation until recently by using ex vivo assays. It was revealed that CLCA1 is important in maintaining the intestinal homeostasis and possess Muc2 cleaving properties, thus playing a role in regulating the mucus dynamics. CLCA1 poses to be an apt candidate to justify modifications in mucus layer upon Rab7 knock-down in mice.

Lack of knowledge regarding the secretion, regulation, and stability of CLCA1 in a cell led us to investigate it at a deeper level. Our data reveals that Rab7 is responsible to direct the CLCA1 containing vesicles towards lysosomes for further degradation. No report has shown the presence of its free molecules in cytoplasm. However, we are not certain whether CLCA1 exclusively localizes in vesicles. It is evident that goblet cells maintain a basal level of CLCA1 to be secreted out in order to maintain the mucus dynamics and homeostasis. Rab7 perturbation disturbs this balance and thus leads to disrupted mucus layer dynamics and sterility, which may initiate inflammation. Together, in the present study we establish the importance of Rab7 in maintaining the intestinal homeostasis by regulating the secretory functioning in the gut. Specifically, the involvement of Rab7 in managing core mucus component CLCA1 opens up avenues for therapeutic interventions. Our findings demonstrate the importance of a fundamental protein like Rab7, which can pose to be the root causative for initiating incurable diseases like IBD.

# Materials and methods

## Key resources table

| Reagent type (species) or resource | Designation | Source or reference | Identifiers | Additional information |
|---|---|---|---|---|
| Cell line (human) | HT29 | ECACC | Lot. 9K003 | |
| Cell line (human) | HT29-MTX-E12 (human) | ECACC | Lot. 8K206 | |
| Cell line (human) | HEK293T Rab7$^{-/-}$ (human) | This paper | | This is a new reagent (details in 'Material and methods') |
| Transfected construct (human) | pSNMUC2-MG plasmid | Prof. Gunnar Hansson **Godl et al., 2002** | | |
| Antibody | Anti-Rab7a (rabbit polyclonal) | Sigma-Aldrich | Cat# R4779 | WB: 1:5000 IHC: 1:400 (indirect) IF: 1:400 (indirect) IEM: 1:1000 |
| Antibody | Anti-Rab7a (mouse monoclonal) | CST | Cat# 95746 | IP: 1:50 IF: 1:400 |
| Antibody | Anti-CLCA1 (rabbit monoclonal) | Abcam | Cat# ab180851 | WB: 1:5000-1:20,000 IHC: 1:100 IF: 1:100 |
| Antibody | Anti-Muc2 (mouse monoclonal) | Santa Cruz | Cat# sc-515032 | IF: 1:200 |
| Antibody | Anti-GAPDH (mouse monoclonal) | Invitrogen | Cat# 39-8600; RRID:AB_2533438 | WB: 1:2000 |
| Antibody | Anti-β-Actin (rabbit polyclonal) | CST | Cat# 4970S | WB: 1:20,000 |
| Antibody | Anti-rabbit HRP (mouse monoclonal) | Jackson ImmunoResearch Laboratories | Cat# 211-032-171; RRID:AB_2339149 | WB: 1:20,000 |
| Antibody | Anti-mouse HRP (goat polyclonal) | Invitrogen | Cat# 31430; RRID:AB_228307 | WB: 1:5000 |
| Antibody | Biotin SP anti-rabbit antibody (goat polyclonal) | Jackson ImmunoResearch Laboratories | Cat# 111-065-003; RRID:AB_2337959 | IF: 1:500 |
| Antibody | Anti-rabbit Alexa Flour 488 (goat polyclonal) | Jackson ImmunoResearch Laboratories | Cat# 111-545-003; RRID:AB_2338046 | IF: 1:400 |
| Antibody | Anti-mouse Alexa Flour 488 (goat polyclonal) | Jackson ImmunoResearch Laboratories | Cat# 115-545-003; RRID:AB_2338840 | IF: 1:400 |
| Antibody | Anti-rabbit Cy5 (goat polyclonal) | Jackson ImmunoResearch Laboratories | Cat# 111-175-144; RRID:AB_2338013 | IF: 1:400 |
| Antibody | Anti-mouse Cy5 (goat polyclonal) | Jackson ImmunoResearch Laboratories | Cat# 111-175-146; RRID:AB_2338713 | IF: 1:400 |
| Antibody | 18 nm Colloidal Gold-AffinityPure Anti-Rabbit (goat polyclonal) | Jackson ImmunoResearch Laboratories | Cat# 111-215-144; RRID:AB_2338017 | IEM: 1:50 |
| Recombinant DNA reagent | pSpCas9(BB)-2A-Puro | addgene | 62988 | |
| Sequence-based reagent | ON-TARGETplus Smart Pool Mouse Rab7a siRNA | Dharmacon | L-040859-02-0005 | |
| Sequence-based reagent | ON-TARGETplus Smart Pool Human Rab7a siRNA | Dharmacon | 010388-00-0005 | |
| Sequence-based reagent | ON-TARGETplus Non-targeting Control Pool | Dharmacon | D-001810-10-20 | |
| Sequence-based reagent | EUB338-Alexa Fluor 647 GCT GCC TCC CGT AGG AGT | IDT | | |
| Peptide, recombinant protein | Streptavidin Cy2 | Jackson ImmunoResearch Laboratories | 16220084 | |

*Continued on next page*

*Continued*

| Reagent type (species) or resource | Designation | Source or reference | Identifiers | Additional information |
|---|---|---|---|---|
| Peptide, recombinant protein | UEA1-FITC | Vector Laboratories | Cat# FL-1061 | IF: 1:400 |
| Peptide, recombinant protein | Phalloidin-594 | Invitrogen | Cat# A12381 | IF: 1:400 |
| Commercial assay or kit | Mouse TNFα ELISA kit | R&D Systems | DY410-05 | |
| Chemical compound, drug | Dextran sodium sulfate | MP Biomedicals | 160110 | |
| Chemical compound, drug | Bafilomycin A1 Ready Made Solution | Sigma-Aldrich | SML1661 | |
| Chemical compound, drug | MG-132 Ready Made Solution | Sigma-Aldrich | M7449 | |
| Other | Gentle cell dissociation reagent | Stemcell Technologies | 100-0485 | For crypt isolation from mice colon ('Materials and methods') |
| Other | Alcian Blue Solution | B8438 | Sigma-Aldrich | For staining mucin gels and mice colon histology ('Materials and methods') |

## Human

Human samples were collected from the All India Institute of Medical Sciences (AIIMS), New Delhi, for non-IBD (IBD suspects), UC and CD groups with age over 18 and below 60 y with inclusion and exclusion criteria specified in *Mustfa et al., 2017*. Two to four biopsies from each patient were collected to be used for qRT-PCR, western blotting, and sectioning. Informed consent forms were acquired from all the patients. Influence of gender was not considered in the present study. UC severity was determined on the basis of UCEIS score. Scores 0–1 were considered under remission, 2–4 as mild, and 5–6 as moderate. Ethics approval for the use of human samples was obtained from both the institutes Regional Centre for Biotechnology (RCB) and AIIMS.

## Mice

C57BL/6 mice were bred and housed in pathogen-free conditions, and provided sterilized food and water at 25°C with 12 hr light/dark cycle in Small Animal Facility of RCB, Faridabad. For DSS colitis experiments, female C57BL/6 mice 6–8 weeks old, weighing 18–20 g were used. Mice were fed with 2.5% DSS (w/v) (36–50 kDa molecular weight; MP Biomedicals) for 3, 5, and 7 d in autoclaved water. Autoclaved water was given to control mice. Body weights were monitored daily, along with colitis symptoms like rectal bleeding and diarrhea. Colon, spleen, liver, and MLN were harvested. Colon and spleen were examined for changes in length and size. Organs were further used for western blots. Distal colons were fixed with 4% paraformaldehyde and processed for sectioning.

## Cell culture and transfection

HT29 (Lot. 09K003) and HT29-MTX-E12 (Lot. 18K206) cell lines were obtained from ECACC. HT29 cells were cultured in RPMI (Merck) containing 10% fetal bovine serum (v/v) (Gibco), 1% sodium pyruvate (Gibco), 2 g/l sodium carbonate (Sigma), and 1% penicillin/streptomycin (Gibco). For differentiating into mucin-secreting goblet-like cells, HT29 cells were grown in glucose-free DMEM (Gibco), supplemented with galactose (250 mM) for 3 d. HT29-MTX-E12 cells were grown in DMEM GlutaMAX medium (Thermo Fisher Scientific) supplemented with 10% fetal bovine serum (v/v), 1% sodium pyruvate, and 1% penicillin/streptomycin. Plasmid transfections and siRNA-mediated knockdowns were done, along with cell seeding using Transfectin reagent (Bio-Rad) and Dharmafect reagent (Horizon Discovery), respectively. To inhibit lysosomal and proteasomal degradation,

cells were treated with 100 nm Bafilomycin (Sigma) and 10 μM MG132 (Sigma) for 6 hr and 8 hr, respectively.

## Intestinal epithelial cells and crypt isolation

Mice colons were flushed with ice-cold PBS to remove luminal content and were opened longitudinally using sterile scissors. Mucus was removed gently using a soft rubber scraper. Colons were treated with 30 mM EDTA for 30 min at 4°C. Cells were scrapped from the colon surface in ice-cold PBS and centrifuged at 2000 rpm for 5 min to collect intestinal epithelial cells for lysate preparation (*Mustfa et al., 2017*). For colonic crypt isolation, intestinal epithelial organoid culture kit protocol (Stemcell Technologies) was followed as per the manufacturer's instructions. Briefly, colons were washed with DPBS (without $Ca^{2+}$ and $Mg^{2+}$) and minced into small pieces. After thorough washing, colon pieces were treated with gentle cell dissociation reagent. Fractions of crypts from solution were collected in 0.1% BSA solution after passing through 70 μm strainer. Crypts were further lysed in RIPA buffer for western blotting.

## CRISPR-Cas9-mediated Rab7 knockout in HEK293T cells

Rab7 knockouts were generated in HEK293T cells using CRISPR-Cas9 technique. sgRNA sequence for human Rab7a was selected (AGGCGTTCCAGACGATTGCA) with AGG as protospacer region (NGG). The gRNA sequence was further cloned into cas9 vector pSpCas9(BB)–2A-Puro (PX459) from addgene. The clone was confirmed by sequencing. $1.3 \times 10^5$ HEK293T cells were seeded and cultured for 24 hr before transfection with Rab7-knockout plasmid using Lipofectamine 2000 (Thermo). Clonal selection of cells was done further by serial dilution as 0.5 cells/well of 96-well plate. Each clone was then tested for Rab7 deletion by western blotting with anti-Rab7 antibody (Sigma).

## Quantitative RT-PCR

Total RNA from the human biopsy samples and mice colon tissue was isolated using RNeasy mini kit (QIAGEN) according to the manufacturer's protocol. cDNA was prepared using 1 μg of total RNA for each sample using cDNA synthesis kit (Bio-Rad). Real-time PCR was performed utilizing SYBR green master mix (Bio-Rad) for 20 μl reaction volume in 96-well plates in CFX96 Real-Time system (Bio-Rad). 18S and HPRT genes were used as housekeeping to normalize the reactions according to the sample origin. The list of primers is in *Supplementary file 2*.

## Western blotting

Human biopsies and mice organ tissues were homogenized using tissue homogenizer (Precellys) in RIPA buffer with added protease arrest (G Biosciences) to prepare protein lysates. Cells were lysed in RIPA buffer (with protease arrest) after PBS wash. Protein amount was measured by BCA solution (Sigma). Equal amount of protein samples was separated on SDS-PAGE gel (12%) electrophoresis and transferred to nitrocellulose membrane (Bio-Rad). Blots were blocked with 5% skim milk for 1 hr at room temperature and probed with antibodies at 4°C overnight against the desired proteins. The primary antibodies used were anti-Rab7 (Sigma, R4779, 1:5000), anti-F (Invitrogen, 39-8600, 1:2000), anti-β-actin (Cell Signaling Technology, 4970S, 1:20,000), anti-CLCA1 (Abcam, ab180851, 1:20,000) (Key Resources Table). Specific secondary antibodies conjugated with HRP were probed for 1 hr at room temperature. Blots were detected and visualized for protein bands with Immobilon Forte western HRP substrate (Millipore) and imaged in Image Quant LAS4000. Band intensities were measured using ImageJ software.

## Histology and immunostaining

Mice distal colons were sectioned (5 μm) after fixing in 4% paraformaldehyde. For histopathology analysis, sections were stained with hematoxylin and counter stained with eosin. Histological scoring of colon sections was evaluated by a blinded pathologist following these parameters: loss of epithelium (0–3), crypt damage (0–3), depletion of goblet cells (0–3), and inflammatory cell infiltrate (0–3). For immunostaining, human biopsy and mice colon samples were sectioned and after fixing with 4% paraformaldehyde antigen retrieval was performed. The sections were probed with anti-Rab7 (Sigma, R4779, 1:400), UEA1-FITC (Vector Laboratories, FL-1061, 1:400), anti-Muc2 (Santa Cruz, sc-515032, 1:200), and anti-CLCA1 (Abcam, ab180851, 1:100) in blocking (5% goat serum) overnight. Sections

were incubated with HRP or fluorophore tagged secondary antibodies for 2 hr. Nucleic acid was stained with DAPI (1 µg/ml). Sections were cured and mounted with Prolong Gold Antifade Reagent (Thermo Fisher). Sections were imaged in confocal microscope (Leica SP8).

## Transmission and immuno electron microscopy

Ultrastructure of HT29-MTX-E12 cells was visualized by transmission electron microscopy (TEM). Cells grown in complete medium were prefixed with Karnovsky fixative (2.5% glutaraldehyde + 2% para-formaldehyde in 0.1 M phosphate buffer, pH 7.4). Post fixation with 1% osmium tetroxide in 0.1 M phosphate buffer for 1 hr, cells were dehydrated in a graded series of ethanol solutions: 30, 50, 70, 80, 90, and 100% and with acetone and toluene. Samples were further embedded in Epon 812 resin and polymerized at 65°C for 48 hr. Ultrathin sections of 70 nm were cut, followed by staining with uranyl acetate and lead acetate and mounted on a grid. Imaging was done with TALOS 200S Transmission electron microscope (Thermo Fisher Scientific) operated at 200 kV. For immuno electron microscopy (IEM), cells were fixed in 0.5% glutaraldehyde in 0.1 M phosphate buffer and were embedded in LR white resin. Thin sections (70 nm) were prepared and incubated for 30 min at room temperature in 1% BSA, followed by overnight incubation with rabbit anti-Rab7 antibody (1:1000 in PBS with 1% BSA) in a humid chamber at 4°C. After washing with 1% BSA, sections were incubated with 18 nm gold affinity pure goat anti-rabbit IgG antibody (Jackson ImmunoResearch Laboratories, 1:50 in 1% BSA) for 2 hr at room temperature. Sections were then contrasted and examined as described above.

## Alcian blue and FISH staining

Mice distal colon pieces containing fecal pellet were fixed in Carnoy's fixative (60% methanol + 30% chloroform + 10% glacial acetic acid), embedded in paraffin and sectioned (5 µm). Sections were dewaxed using Histoclear (Sigma), stained with Alcian blue solution, and counterstained with nuclear fast red (*Musch et al., 2013*). Sections were visualized in Nikon bright-field microscope and mucus thickness was measured using ImageJ software. For FISH, dewaxed sections were incubated with 1.3 µg of general bacterial probe EUB338-Alexa Fluor 647 overnight at 50°C in a humid chamber (*Swidsinski et al., 2005*). Nucleus was stained with DAPI (1 µg/ml). Sections were imaged in confocal microscope (Leica SP8). Bacteria in mucus layer per section were calculated using particle counter in ImageJ software.

## Immunocytochemistry and SIM

Cells were seeded on 18 mm glass coverslips, fixed with 4% methanol-free paraformaldehyde, and blocked in 0.1% BSA + 0.01% TritonX-100 for 1 hr. Cells were probed with anti-CLCA1 (Abcam, ab180851, 1:100), anti-Muc2 (Santa Cruz, sc-515032, 1:200), anti-Rab7 (Sigma, R4779, 1:400), and Phalloidin-594 (Invitrogen, A12381, 1:400) at 4°C overnight. Cells were further incubated with fluoro-phore tagged secondary antibodies for 2 hr. Nucleic acid was stained with DAPI (1 µg/ml). Coverslips were mounted with Prolong Gold Antifade Reagent (Thermo Fisher) and visualized in confocal micro-scope (Leica SP8) for immunocytochemistry and in Elyra PS1 (Carl Zeiss) for structured illumination microscopy.

## Polymer-based knockdown in mice

Female C57BL/6 mice (aged 6–8 wk, weight 18–23 g) were divided into six groups. Group 1 was untreated control group. Group 2 mice were treated with control nanogels only. Group 3 mice were treated with *Rab7* siRNA nanogels. Group 4 mice were fed with only ~2.5% DSS. Group 5 mice were fed with ~2.5% DSS and treated with control nanogels. Group 6 mice were fed with ~2.5% DSS and treated with *Rab7* siRNA-nanogels. For one mouse, *Rab7* siRNA nanogels were prepared by incu-bating TAC6 (100 µl of stock 2 mg/ml) with siRNA (0.6 µg/per dose) for 30 min, followed by incubation with 10 µl of SPA (2 mg/ml) for 10 min. For control nanogels, scrambled siRNA was fed.

## Mucus isolation and separation

Mice colons were harvested from each mouse and after washing with ice-cold PBS were cut open longi-tudinally. Mucus was scraped using rubber scraper, resuspended in PBS + protease inhibitor cocktail (2×), and snap frozen. Further, samples were denatured in reducing buffer at 95°C for 20 min followed by additional reduction at 37°C for 1 hr. Composite gradient agarose-PAGE gel was prepared with

0.5–1% agarose, 0–6% polyacrylamide, and 0–10% glycerol and solidified at 37°C for 1 hr followed by overnight incubation in a humid chamber. Mucus samples were separated on mucin gel electrophoresis and stained with Alcian blue to visualize highly glycosylated mucins (*Johansson et al., 2009*).

For isolation of mucus from human samples, the biopsy tissue was treated with N-acetyl cysteine (2% PBS solution) for 5 min at 4°C. The supernatant was collected and centrifuged at 2000 rpm for 5 min to remove cell debris. Protease inhibitor (2×) was added to the supernatant. The mucus samples were further denatured in reducing buffer at 95°C for 15 min.

## Mucus proteomics by LC MS/MS

LC-MS/MS analysis was performed using mucus sample lysates. In-Gel digestion was carried out using Trypsin gold (Promega). The samples were purified using C18 SepPak columns (Thermo, USA). The peptide samples were dissolved in 98% milliQ-H$_2$O, 2% acetonitrile, 0.1% formic acid. Tandem MS analysis was performed using a 5600 TripleTOF analyzer (ABSCIEX) in Information-Dependent Mode. Protein identification was performed with MaxQuant software (v1.6.0.16) under default settings, except for the peptide length that was set from 6 to 40. Masses were searched against the Mouse UniProt Proteome (downloaded April 27, 2020) with additional mice mucin database, VerSeDa, antimicrobial peptides (from UniProt), and cytokines (from UniProt). The peptides resulting from MaxQuant were then processed as follows: known MS contaminants and reverse sequences were filtered out; we allowed peptides that had at least two valid LFQ intensities out of sample replicates and included razor peptides, which belong to a unique MaxQuant 'protein group'; finally, missing values were imputed to a random value selected from a normal distribution of 0.3 s.d. and downshifted 1.8 s.d. The data processing was performed using Python v3.8. Statistical analyses were performed using R v4.0.2. In R data was visualized using the complexheatmap and ggplot packages (*Gu et al., 2016*; *Wickham, 2009*). Protein annotation and gene ontology analysis were performed with the gene-set enrichment analysis using the enrichR webtool (*Chen et al., 2013*; *Kuleshov et al., 2016*; *Xie et al., 2021*).

## 16S metagenomic profiling

Fecal pellets were collected from each mouse and submitted to miBiome Therapeutics for processing and analysis. Bacterial genomic DNA was isolated by QIAamp PowerFecal Pro DNA Kit (QIAGEN, 51804). DNA samples were quality checked and subjected to library preparation in alignment with the 16S metagenomic library preparation protocol from Illumina Inc to target the 16S V3 and V4 region. The full-length primer sequences targeting the V3-V4 region are 16S Amplicon PCR Forward Primer 5'TCGTCGGCAGCGTCAGATGTGTATAAGAGACAGCCTACGGGNGGCWGCAG and 16S Amplicon PCR Reverse Primer 5'GTCTCGTGGGCTCGGAGATGTGTATAAGAGACAGGACTACHVGGG-TATCTAAC. Libraries were diluted to 4 nM, pooled, spiked with 20% PhiX pre-made library from Illumina, and loaded on a MiSeq v3 kit. Sequencing was performed for 2 × 300 cycles. The original raw data obtained from Illumina platform are recorded in FASTQ files, which contains sequence information (paired end reads) and corresponding sequencing quality information (Q score). The reads obtained from the instrument were made adapter free, using the adapter trimming of local run manager on Illumina. The paired-end reads were assembled, filtered, trimmed, and aligned to SILVA 16S database using mother software v.1.44.1. The reads were clustered based on similarity and further clustered into OTUs using mothur software v.1.44.1 at 97% identity against database greengenes_13_8_99.

## ELISA

Colonic mucus was scrapped and diluted in PBS. The homogenate was centrifuged to remove debris and the supernatant was transferred to a new tube. The samples were quantified by BCA assay and further equal protein was assayed by ELISA for TNF-α using the manufacturer's protocol (R&D Systems, USA).

## Statistical analysis

Results were analyzed and plotted using GraphPad prism 8.0.1 software. All results were expressed as mean standard error from individual experiment done in triplicates. Data were analyzed with standard unpaired two-tailed Student's *t*-test and the Welch's *t*-test where applicable. $p < 0.05$ were considered as statistically significant.

## Acknowledgements

We thank the mass spectrometry facility, the Central Instrumentation Facility (CIF) of Regional Centre for Biotechnology (RCB), and Small Animal facility at NCR, Biotech Science Cluster. We also thank Sophisticated Analytical Instrumentation Facility, AIIMS (SAIF-AIIMS) and Advanced Technology Platform Centre (ATPC) for imaging facilities. We are thankful to Professor Gunnar C Hansson for Muc2 plasmids and Dr. Amit Tuli for Rab7 plasmids. We acknowledge Addgene for the plasmid. We thank Dr. Bobby Cherayil for all the discussions and feedback during the manuscript preparation. This work was supported by DBT grant (BT/PR45284/CMD/150/9/2022), RCB Core Grant, ICMR-SRF award of PG, CSIR-JRF programme (Council of Scientific and Industrial Research, Govt. of India) of PG.

## Additional information

### Funding

| Funder | Grant reference number | Author |
| --- | --- | --- |
| Department of Biotechnology, Ministry of Science and Technology, India | BT/PR45284/ CMD/150/9/2022 | Preksha Gaur Chittur Srikanth |

The funders had no role in study design, data collection and interpretation, or the decision to submit the work for publication.

### Author contributions

Preksha Gaur, Data curation, Investigation, Methodology, Writing - original draft, Project administration, Writing - review and editing; Yesheswini Rajendran, Bhagyashree Srivastava, Data curation, Investigation; Manasvini Markandey, Shikha Chaudhary, Shaifali Tyagi, Resources; Vered Fishbain-Yoskovitz, Formal analysis; Gayatree Mohapatra, Yifat Merbl, Supervision, Writing - review and editing; Aamir Suhail, Subhash Chandra Yadav, Amit Kumar Pandey, Supervision; Avinash Bajaj, Vineet Ahuja, Resources, Supervision; Chittur Srikanth, Conceptualization, Methodology, Writing - original draft, Writing - review and editing

### Author ORCIDs

Preksha Gaur ⓘ http://orcid.org/0000-0002-8011-1821
Avinash Bajaj ⓘ http://orcid.org/0000-0002-1333-9316
Chittur Srikanth ⓘ http://orcid.org/0000-0001-9381-1213

### Ethics

Informed consent form was obtained from all the patients and the study protocol was submitted to the ethics committee of the institutes: RCB (IEC-141/05.03.2021, RP-34/2021) and AIIMS (IEC/NP-18 9/2013&RP-12/17.06.201307.06.2013).
Animals ethics proposal was approved by the RCB Institutional Animal Ethics Committee (approval no. RCB/IAEC/2020/083).

Reviewer #1 (Public Review): https://doi.org/10.7554/eLife.89776.3.sa1
Reviewer #2 (Public Review): https://doi.org/10.7554/eLife.89776.3.sa2
Author response https://doi.org/10.7554/eLife.89776.3.sa3

## Additional files

### Supplementary files

- Supplementary file 1. List of patient clinical parameters.
- Supplementary file 2. List of qRT-PCR primers.
- MDAR checklist

## Data availability

All the numerical source data generated during the study is made available as attached supporting excel files for Metagenomics (*Figure 5—source data 1*) and Mucus proteomics (*Figure 6—source data 1*).

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
