## [Editor Report · eLife assessment]

This is an **important** study for understanding the pathogenesis of ulcerative colitis. It **convincingly** demonstrates reduced levels of the vesicular trafficking protein Rab7 in ulcerative colitis and Crohn's disease, leading to altered levels of calcium-activated chloride channel regulator 1 (CLCA1) and subsequent mucin dysregulation, highlighting Rab7's significance in gut homeostasis maintenance. The article advances the field as it provides insights into a novel regulatory pathway implicated in ulcerative colitis, potentially paving the way for the development of targeted therapeutic interventions.

---

## [Referee Report · Reviewer #1 (Public Review)]

Assessment:

The manuscript titled 'Rab7 dependent regulation of goblet cell protein CLCA1 modulates gastrointestinal 1 homeostasis' by Gaur et al discusses the role of Rab7 in the development of ulcerative colitis by regulating the lysosomal degradation of Clca1, a mucin protease. The manuscript presents interesting data, and provides a potential molecular mechanism for the pathological alterations observed in ulcerative colitis.

Strengths:

The manuscript used a multi-pronged approach and compares patient samples, mouse models of DSS and protocols that allow differentiation of goblet cells. They also use a nanogel-based delivery system for siRNAs, which is ideal for knockdown of specific genes in the gut.

Weaknesses:

The manuscript should also mention the limitations of the study.

---

## [Referee Report · Reviewer #2 (Public Review)]

Summary:

In this work, the authors report a role for the well-studied GTPase Rab7 in gut homeostasis. The study combines cell culture experiments with mouse models and human ulcerative colitis patient tissues to propose a model where, Rab7 by delivering a key mucous component CLCA1 to lysosomes, regulates its secretion in the goblet cells. This is important for the maintenance of mucous permeability and gut microbiota composition. In the absence of Rab7, CLCA1 protein levels are higher in tissues as well as the mucus layer, corroborating with the anti-correlation of Rab7 (reduced) and CLCA1 (increased) from ulcerative colitis patients. The authors conclude that Rab7 maintains CLCA1 level by controlling its lysosomal degradation, thereby playing a vital role in mucous composition, colon integrity, and gut homeostasis.

Strengths:

The biggest strength of this manuscript is the combination of cell culture, mouse model, and human tissues. The experiments are largely well done and in most cases, the results support their conclusions. The authors go to substantial lengths to find a link, such as alteration in microbiota, or mucus proteomics.

Weaknesses:

There are also some weaknesses that need to be addressed. The association of Rab7 with UC in both mice and humans is clear, however, claims on the underlying mechanisms are less clear. Does Rab7 regulate specifically CLCA1 delivery to lysosomes, or is it an outcome of a generic trafficking defect? CLCA1 is a secretory protein, how does it get routed to lysosomes, i.e. through Golgi-derived vesicles, or by endocytosis of mucous components? Mechanistic details on how CLCA1 is routed to lysosomes will add substantial value.

Why does the level of Rab7 fluctuate during DSS treatment (Fig 1B)? Does the reduction seen in Rab7 levels (by WB) also reflect in reduced Rab7 endosome numbers? Are other late endosomal (and lysosomal) populations also reduced upon DSS treatment and UC? Is there a general defect in lysosomal function?

While it is clear that the pattern of Muc2 in WT and Rab7-/- cells are different, how this corroborates with the in vivo data on alterations in mucus layer permeability - as claimed - is not clear.

The use of an in vivo intestine-specific Rab7 silencing model is good. Why does Rab7 KD itself not capitulate aspects of DSS treatment, rather it seems to exacerbate it.

The use of mucous proteomics to identify mechanisms of Rab7-mediated phenotype is a good approach. The replicates in the proteomics dataset (Fig 6F) do not seem to match. Detailing of methodology used for analysis will help to overcome these doubts.

The work shows a role for a well-studied GTPase, Rab7, in gut homeostasis. This is an important finding and could provide scope and testable hypotheses for future studies aimed at understanding in detail the mechanisms involved.

---

## [Author Response]

The following is the authors’ response to the original reviews.

**Reviewer #1 (Public Review):**
Assessment:The manuscript titled 'Rab7 dependent regulation of goblet cell protein CLCA1 modulates gastrointestinal 1 homeostasis' by Gaur et al discusses the role of Rab7 in the development of ulcerative colitis by regulating the lysosomal degradation of Clca1, a mucin protease. The manuscript presents interesting data and provides a potential molecular mechanism for the pathological alterations observed in ulcerative colitis. Gaur et al demonstrate that Rab7 levels are lowered in UC and CD. However, a similar analysis of Rab7 levels in ulcerative colitis (UC) and Crohn's disease (CD) patient samples was conducted recently (Du et al, Dev Cell, 2020) which showed that Rab7 levels are found to be elevated under these conditions. While Gaur et al have briefly mentioned Du et al's paper in passing in the discussion, they need to discuss these contradictory results in their paper and clarify these differences. Additionally, Du et al are not included in the list of references.Strengths:The manuscript used a multi-pronged approach and compares patient samples, mouse models of DSS, and protocols that allow differentiation of goblet cells. They also use a nanogel-based delivery system for siRNAs, which is ideal for the knockdown of specific genes in the gut.Weaknesses:(1) Du et al, Dev Cell 2020 (https://doi.org/10.1016/j.devcel.2020.03.002) have previously shown that Rab7 levels are elevated in a similar set of colonic samples (age group, number etc.) from UC and CD patients. Gaur et al have not discussed this paper or its findings in detail, which directly contradicts their results. Clarification regarding this should be provided.

We thank and appreciate the reviewer for bringing this point.

The results shown by Du et al, Dev Cell, 2020 depict elevated expression of Rab7 in UC and CD patients compared to controls. In first occurrence, these results appear contradictory, but there may be a few possible explanations for this.

Firstly, Rab7 expression levels may fluctuate in the tissue depending on the degree of the gut inflammation. This can be concluded from our observations in DSS-mice dynamics model and the human patient samples with mild and moderate UC. Furthermore, Du et al provide no information of the severity of the condition among the patients employed in the study. Our motive, in the current work, was to emphasize this aspect. This point was mentioned in the discussion section of the manuscript. However, in view of the reviewer’s concern, we have now added a detailed comment on this in the main text of the revised version of the manuscript.

Secondly, the control biopsies in our investigation were acquired from non-IBD patients, and not what was done by Du et al., wherein biopsies from the normal para-carcinoma region of the colorectal cancer patients were used. One cannot overlook the fact that physiological and molecular changes are apparent even in non-inflamed regions in the gut of an IBD or CRC patient. It is possible that the observed discrepancy arises due to the differences in the sample type used for comparing the Rab7 expression.

Finally, the main sub-tissue region showing a decrease in Rab7 expression in UC samples, appeared to be the Goblet cells which was not covered by Du et al.

Keeping these points in mind we do not think that there is a contradiction in our findings with that of Du et al., 2020. In the revised submission some of these explanations are incorporated (Lines 106-109).

This was an oversight from our side. We have actually mentioned Du et al., 2020 in the discussion (line number 345) but somehow the reference was missing in the main list. We have ensured that the reference is included in the revised version and that their findings are included both in main text and in the discussion.

**Reviewer #2 (Public Review):**
Summary:In this work, the authors report a role for the well-studied GTPase Rab7 in gut homeostasis. The study combines cell culture experiments with mouse models and human ulcerative colitis patient tissues to propose a model where, Rab7 by delivering a key mucous component CLCA1 to lysosomes, regulates its secretion in the goblet cells. This is important for the maintenance of mucous permeability and gut microbiota composition. In the absence of Rab7, CLCA1 protein levels are higher in tissues as well as the mucus layer, corroborating with the anticorrelation of Rab7 (reduced) and CLCA1 (increased) from ulcerative colitis patients. The authors conclude that Rab7 maintains CLCA1 level by controlling its lysosomal degradation, thereby playing a vital role in mucous composition, colon integrity, and gut homeostasis.Strengths:The biggest strength of this manuscript is the combination of cell culture, mouse model, and human tissues. The experiments are largely well done and, in most cases, the results support their conclusions. The authors go to substantial lengths to find a link, such as alteration in microbiota, or mucus proteomics.Weaknesses:(1) There are also some weaknesses that need to be addressed. The association of Rab7 with UC in both mice and humans is clear, however, claims on the underlying mechanisms are less clear. Does Rab7 regulate specifically CLCA1 delivery to lysosomes, or is it an outcome of a generic trafficking defect?

We thank the reviewer for the insightful comment. We would like to bring forth the following explanation for each these concerns:

Our immunofluorescence imaging experiments revealed co-localization of Rab7 protein with CLCA1 and the lysosomes (Fig 7I). In addition, the absence of Rab7 affects the transport of CLCA1 to lysosomes (Fig 7J). This demonstrates that Rab7 may be involved in regulation of CLCA1 transport (presumably along with other cargo), to lysosomes selectively. However, we do recognize that the point raised by the reviewer about possible effect of a generic trafficking defect is valid.

(2) CLCA1 is a secretory protein, how does it get routed to lysosomes, i.e., through Golgi-derived vesicles, or by endocytosis of mucous components? Mechanistic details on how CLCA1 is routed to lysosomes will add substantial value.

As mentioned in the manuscript, the trafficking of CLCA1 protein or CLCA1-containing vesicles within the goblet cell is unknown, with no information on the proteins involved in its mobility. The switching of CLCA1 containing vesicles from the secretory route to lysosomes needs extensive investigation involving overall trafficking of the protein. Taken together, the complete answer to both these important questions will need a series of experiments and those may be interesting avenues for future research.

(3) Why does the level of Rab7 fluctuate during DSS treatment (Fig 1B)?

This is a very thoughtful point from the reviewer. We detected a distinct pattern of Rab7 expression fluctuation in intestinal epithelial cells after DSS-dynamics treatment in mice. Perhaps, these changes are the result of complex cellular signaling in response to the DSS treatment. Rab7, being a fundamental protein involved in protein sorting pathway, is expected to undergo alteration based on cells requirement. Presently there are no reports suggesting the regulatory mechanisms that govern Rab7 levels in the gut.

(4) Does the reduction seen in Rab7 levels (by WB) also reflect in reduced Rab7 endosome numbers?

We observed reduction in Rab7 expression both at RNA and protein levels. To confirm whether this alteration will lead to reduced Rab7 positive endosome numbers may require detailed investigations.

(5) Are other late endosomal (and lysosomal) populations also reduced upon DSS treatment and UC? Is there a general defect in lysosomal function?

There are no direct evidences showing reduction in the late endosomal and lysosomal population during gut inflammation, but few studies link lysosomal dysfunction with risk for colitis (doi:10.1016/j.immuni.2016.05.007).

(6) The evidence for lysosomal delivery of CLCA1 (Fig 7 I, J) is weak. Although used sometimes in combination with antibodies, lysotracker red is not well compatible with permeabilization and immunofluorescence staining. The authors can substantiate this result further using lysosomal antibodies such as Lamp1 and Lamp2. For Fig 7J, it will be good to see a reduction in Rab7 levels upon KD in the same cell.

We used Lysotracker red in live cells followed by fixation. So, permeabilization issues were resolved. Lamp1, as suggested by the reviewer, is definitely a better marker for lysosomes in immunofluorescence studies, but is also shown to mark late endosomes (doi: 10.1083/jcb.132.4.565). As Rab7 protein also marks the late endosomes, using Lamp1 may leave the ambiguity of CLCA1 in Rab7 positive late endosomes versus lysosomes. Nevertheless, we have carried out this experiment, as suggested by the reviewer, by staining the cells with LAMP1 (author response image 1). As demonstrated in our previous data, the colocalization of CLCA1 with LAMP1 positive vesicles decreased upon Rab7 knockdown. Also, we observed a decrease in the intensity of LAMP1 staining in cells with Rab7 knockdown. Additionally, we noted a reduction in the LAMP1 staining intensity in cells where Rab7 was knocked down. This observation can be attributed to the decrease in the presence of Rab7-positive vesicles or late endosomes which also exhibit LAMP1 staining.

**Author response image 1. sa3fig1:** (**A**) Representative confocal images of HT29-MTX-E12 cells transfected with either scrambled siRNA (control) or Rab7 siRNA (Rab7Knockdown). Cells are stained with CLCA1 (green) using antiCLCA1 antibody and lysosomes with LAMP1. (**B**) Graph shows quantitation of colocalization between CLCA1 and LAMP1 from images (n = 20) using Mander’s overlap coefficient. Inset shows zoomed areas of the image with colocalization puncta (yellow) marked with arrows.

(7) In this connection, Fig S3D is somewhat confusing. While it is clear that the pattern of Muc2 in WT and Rab7-/- cells are different, how this corroborates with the in vivo data on alterations in mucus layer permeability -- as claimed -- is not clear.

The data in Fig. S3D suggest the involvement of Rab7 in packaging of Muc2. The whole idea for doing this experiment was to support our observation in the Rab7KD-mice model where mucus layer was seen to be loose and more permeable in Rab7 deficient mice.

(8) Overall, the work shows a role for a well-studied GTPase, Rab7, in gut homeostasis. This is an important finding and could provide scope and testable hypotheses for future studies aimed at understanding in detail the mechanisms involved.

We thank the reviewer for this comment.

**Recommendations for the authors:**

**Reviewer #1 (Recommendations For The Authors):**
Specific questions to the authors:(1) Why is the dotted line in Fig. 1c at -7.5? What does this signify?

Response: The dotted line was intended to represent the baseline; in the revised manuscript it is corrected and placed at y=0.

(2) Du et al should be cited. Fig 6 K-Q from Du et al should be discussed and reasons for contradictory findings should be given in greater detail, rather than a single sentence in the discussion.

Response: The reference for Du et al is included in the list and the possible reasons the findings of the current work are discussed in the main text (Line 106-109).

(3) Fig1. Why are Rab7 levels low even in remission patient samples? Can DSS be withdrawn to induce remission followed by analysis of colonic samples?

Response: A possible explanation for this observation could be that the restoration of Rab7 levels may not immediately follow the resolution of clinical symptoms in remission patients. After the remission initiation, the normalization of cellular processes, including the regulation of Rab7 expression, might exhibit a time lag. A thorough investigation of Rab7 levels and the allied pathways at different time points during the remission phase could provide deeper insights into the gradual dynamics of recovery. As suggested by the reviewer, DSS withdrawal induced recovery model can be utilized for understanding the same and could be a good approach for future investigations.

(4) Fig. 2: Single-channel fluorescence should be shown.

Response: The single channel fluorescence images are incorporated in Fig. S2.

(5) Line 456 should be modified. 'Blind pathologist' does not read well!

Response: The line has been modified with ‘Blinded pathologist’.

(6) Other inflammatory markers, cytokine levels should be looked at in addition to TNF alpha.

Response: TNF-α is a crucial mediator in intestinal inflammation, actively contributing to the development of IBD. Elevated levels of TNF-α are observed in patients of IBD (Billmeier U. et al, World J Gastroenterol. 2016). In the current work, while probing for TNF-α our primary objective was to examine this significant indicator of colitis following Rab7 knockdown in mice, aiming to gain insights into heightened gut inflammation.

(7) Quantitation of S3D should be provided.

Response: The dispersed expression of Muc2 was observed in n = 20 cells per sample and it was a qualitative observation. The aim was to identify any changes in Muc2 packaging under Rab7 knockout conditions.

(8) Microbiota analysis should include Rab7KD+DSS mice.

Response: We understand the importance of this point, however, in the current work our primary objective was to specifically investigate changes in microbial diversity and abundance in Rab7KD mice compared to both DSS+CScr and CScr mice. Rab7KD+DSS mice is expected to show higher dysbiosis in comparison to DSS+CScr.

(9) Fig 6 H and I, G. How do Clca1 levels reduce in Rab7kd +DSS relative to Scr+DSS while they are higher in Rab7kd compared to Scr. Comment.

Response: The decreased expression of CLCA1 in the mucus of DSS+Rab7KD mice can be attributed to a consequence of significant reduction in goblet cell numbers in these mice, as evidenced by the observed loss of these cells (Fig.S3 B and Fig. S3C). CLCA1 is exclusively secreted by goblet cells, so a decline in their numbers directly affects CLCA1 levels.

(10) How are Rab7 levels downregulated? What is the predicted mechanism?

Response: While our current study didn't explore this aspect, it's worth noting that Rab7 protein levels undergo regulation through various mechanisms, including post-translational modifications such as Ubiquitination and SUMOylation. These modifications are known to regulate Rab7 stability, transport and recycling. Specific experiments conducted during this study (work not included in the manuscript) indicated the participation of SENP7, a deSUMOylase, in controlling the stability of Rab7 protein, particularly in the context of colitis.Additionally, goblet cell specific mechanisms are also likely to be controlling the Rab7 in the gut.

(11) What is the explanation for opposite changes in CLCa1 RNA (down) and protein (up).

Response: The reduction in CLCA1 at the RNA level could be associated with the decrease in goblet cell numbers during colitis. Our investigation indicates that Rab7 predominantly influences CLCA1 at the protein level by impacting its degradation pathway. It is important to acknowledge that not all the alterations in CLCA1 observed during colitis can be solely attributed to Rab7, but our study has identified a connection between Rab7 and CLCA1.

(12) In light of Du et al, it would be interesting to see how the number of peroxisomes changes upon alteration of Rab7 levels.

Response: The suggestion by the reviewer is noteworthy. Since, being an altogether different domain, it deviates from the primary objectives of current work. Here, our goal was specifically on exploring the role of Rab7 in goblet cell functioning. Thus is an attractive theme for future investigations.

(13) While Gaur et al suggest in their discussion that Du et al may have observed an upregulation in Rab7 levels in different cell types of the intestine, this is not apparent from the data provided. Tissue sections should be carefully analysed to provide data supporting this observation. Differences in reagents used (antibodies) should also be considered. As far as the human patient data is concerned, it does not appear that the sample stages are very different across the two manuscripts (based on age, inclusion criteria etc.).

Response: This has been explained in detail in our public comments.

**Reviewer #2 (Recommendations For The Authors):**
(1) In general, image-based measurements could be done better (for example, object-based statistics than pixel-based overlaps) and represented differently. It is difficult to appreciate the reduction in Rab7 levels in goblet cells in Fig 2 A, C. It might be good to show the channels separately, and perhaps use an intensity gradient LUT for the Rab7 channel.

Response: The single channel fluorescence images are incorporated in Fig. S2.

(2) The EM images, and particularly Fig 2F are not convincing, with an oddly square-shaped vesicle. I'm not sure what value they are adding to the interpretation.

Response: The observed square-shaped vesicle in Fig. 2F could be attributed to the dynamic nature of vesicles within a cell. This dynamicity allows them to adopt various shapes depending on their state and function within the cell. The presence of Rab7 near vacuoles of goblet cells signify its probable involvement in the regulation of secretory function of these cells which is the key aspect being covered in this work.

(3) A general method question concerns the definition of the distal colon. How is this decided, particularly when colon lengths are reduced upon DSS treatment?

Response: The murine colon is divided into proximal and distal colon of mouse and has a visual difference of inner folds which are quite prominent in proximal colon. Additionally, the portion towards the rectum (predominantly distal colon) was majorly utilized for the experiments. In each case the various experimental groups were matched for the respective areas.

(4) The use of an in vivo intestine-specific Rab7 silencing model is good. Why does Rab7 KD itself not capitulate aspects of DSS treatment, rather it seems to exacerbate it.

Response: Our objective was to determine whether the downregulation of Rab7 during colitis was the cause or consequence of gut inflammation. Interestingly, our investigation using the murine Rab7 knockdown model revealed that the reduction of Rab7 expression in the intestine exacerbates inflammation. Subsequent analysis demonstrated that the absence of Rab7 disrupts goblet cell secretory function, consequently contributing to heightened inflammation. Our findings overall suggest that Rab7 downregulation is not merely a consequence but plays a contributory role in aggravating inflammation in the context of colitis.

(5) The axes labels in Fig 5 are not readable. It is unclear how Rab7 KD is more similar in gut microbiota phenotypes to DSS than to CScr.

Response: The microbial analysis revealed an abnormal composition of gut microbiota in Rab7KD mice compared to CScr. Interestingly, this composition exhibited some similarity to the inflamed gut microbiota observed in DSSScr mice. The analysis further demonstrated a shift in microbial diversity in Rab7KD mice, showcasing characteristics akin to those observed in inflamed mice. This similarity in gut microbiota phenotypes between Rab7KD and DSSScr suggests a potential link or influence of Rab7 downregulation on the microbiota, contributing to the observed similarities with DSS-induced inflammation.

(6) The use of mucous proteomics to identify mechanisms of Rab7-mediated phenotype is a good approach. The replicates in the proteomics dataset (Fig 6F) do not seem to match. Detailing of methodology used for analysis will help to overcome these doubts.

Response: The identified proteins in different samples of mucus proteomics were subjected to label free quantification. Subsequently, the significantly altered proteins were subjected to analysis with the False Discovery Rate (FDR) to control for potential false positives and ascertain the validity of the findings.

(7) It will be good to see the immunoblots showing the negative correlation between Rab7 and CLCL1 in Fig 7D.

Response: Fig. 7C shows western blot for protein expression of CLCA1of the same control and UC samples which were used in Fig. 1F to show Rab7 expression. Fig. 7D is the quantitative correlation plot for Fig. 1F (Rab7 expression) and Fig. 7C (CLCA1 expression).

(8) Why is UC different from the DSS model for Rab7 gene expression but not protein levels? Endosomal counts could help address this.

Response: We encountered challenges in accurately counting the individual puncta of Rab7 expression in immunofluorescence images due to the nature of tissue samples. Locating endosomes within a single cell proved to be challenging, and the proximity of many puncta made it difficult to delineate them individually. Despite these technical difficulties, the intriguing prospect of correlating Rab7 expression with endosomal counts remains a compelling aspect that may well be area for future investigations.